**Investigation of satellite vertical sensitivity on long-term retrieved lower tropospheric ozone trends**
Richard J. Pope[1,2], Fiona M. O'Connor[3,4], Mohit Dalvi[3], Brian J. Kerridge[5,6], Richard Siddans[5,6], Barry G.
Latter[5,6], Brice Barret[7], Eric Le Flochmoen[7], Anne Boynard[8,9] , Martyn P. Chipperfield[1,2], Wuhu Feng[1,10],
Matilda A. Pimlott[1], Sandip S. Dhomse[1,2], Christian Retscher[11], Catherine Wespes[12] and Richard Rigby[1,13]
*1: School of Earth and Environment, University of Leeds, Leeds, United Kingdom*
*2: National Centre for Earth Observation, University of Leeds, Leeds, United Kingdom*
*3: Met Office Hadley Centre, Exeter, United Kingdom*
*4: Department of Mathematics & Statistics, Global Systems Institute, University of Exeter, United Kingdom*
*5: Remote Sensing Group, STFC Rutherford Appleton Laboratory, Chilton, United Kingdom*
*6: National Centre for Earth Observation, STFC Rutherford Appleton Laboratory, Chilton, United Kingdom*
*7: Laboratoire d'Aérologie/OMP, Université de Toulouse, Toulouse, France*
*8: LATMOS/IPSL, Sorbonne Université, UVSQ, CNRS, Paris, 75005, France*
*9: SPASCIA, Ramonville-Saint-Agne, 31520, France*
*10: National Centre for Atmospheric Science, University of Leeds, Leeds, United Kingdom*
*11: European Space Agency, ESRIN, Frascati, Italy*
*12: Université libre de Bruxelles (ULB), Spectroscopy, Quantum Chemistry and Atmospheric Remote Sensing,*
*Brussels, Belgium*
*13: Centre for Environmental Modelling and Computation, University of Leeds, Leeds, United Kingdom*
Submitted for *Atmospheric Chemistry and Physics*
*Correspondence to*: Richard J. Pope (r.j.pope@leeds.ac.uk)
**Key Points**
• Satellite lower tropospheric column ozone (LTCO$_3$) records in the northern hemisphere show small
trends with large uncertainty ranges between 2008 and 2017.
• Modelled LTCO$_3$ over that period is temporally stable and application of the satellite averaging
kernels (AKs), accounting for the satellite vertical sensitivity, to the model yields little impact on the
simulated trends.
**Abstract:**
Ozone is a potent air pollutant in the lower troposphere and an important short-lived climate forcer (SLCF) in
the upper troposphere. Studies investigating long-term trends in tropospheric column ozone (TCO$_3$) have
shown large-scale spatiotemporal inconsistencies. Here, we investigate the long-term trends in lower
tropospheric column ozone (LTCO$_3$, surface-450 hPa sub-column) by exploiting a synergy of satellite and
ozonesonde datasets and an Earth System Model (UKESM) over North America, Europe and East Asia for the
decade 2008-2017. Overall, we typically find small LTCO$_3$ linear trends with large uncertainty ranges from the
Ozone Monitoring Instrument (OMI) and the Infrared Atmospheric Sounding Interferometer (IASI), while
model simulations indicate a stable LTCO$_3$ tendency. The satellite apriori datasets show negligible trends
indicating that any year-to-year changes in spatiotemporal sampling of these satellite data sets, over the
period concerned, has not artificially influenced their LTCO$_3$ temporal evolution. The application of the
satellite averaging kernels (AKs) to the UKESM simulated ozone profiles, accounting for the satellite vertical
sensitivity and allowing for like-for-like comparisons, has a limited impact on the modelled LTCO$_3$ tendency
in most cases. While, in relative terms, this is more substantial (e.g. in the order of 100%), the absolute
magnitudes of the model trends show negligible change. However, as the model has a near-zero tendency,
artificial trends were imposed on the model time-series (i.e. LTCO$_3$ values rearranged from smallest to
largest) to test the influence of the AKs but simulated LTCO$_3$ trends remained small. Therefore, the LTCO$_3$
tendency between 2008 and 2017 in northern hemispheric regions are likely small, with large uncertainties,
and it is difficult to detect any small underlying linear trends due to inter-annual variability or other factors
which require further investigation (e.g. the radiative transfer scheme (RTS) used and/or the inputs (e.g.
meteorological fields) used in the RTS).

## 1. Introduction

Tropospheric ozone (TO$_3$) is a short-lived climate forcer (SLCF) and an important greenhouse gas (GHG;
Myhre et al., 2013; Forster et al., 2021). TO$_3$ is also a hazardous air pollutant with adverse impacts on human
health (Doherty et al., 2017; WHO, 2022) and agricultural/natural vegetation (Sitch et al., 2007; Hollaway et
al., 2012). Since the pre-industrial (PI) period, anthropogenic activities have increased the atmospheric
loading of ozone (O$_3$) precursor gases, most notably methane (CH$_4$) and nitrogen oxides (NO$_x$) resulting in an
increase in TO$_3$ of 25-50% since 1900 (Gauss et al., 2006; Lamarque et al., 2010; Young et al., 2013). The PI to
present day (PD) radiative forcing (RF) from TO$_3$ is estimated by the Intergovernmental Panel on Climate
Change (IPCC) to be 0.47 Wm$^{-2}$ (Forster et al., 2021) with an uncertainty range of 0.24-0.70 Wm$^{-2}$.
During the satellite-era (i.e. since the mid-1990s), extensive records of TO$_3$ have been produced, e.g. by the
European Space Agency Climate Change Initiative (ESA-CCI; ESA, 2019). However, the large presence of
stratospheric O$_3$, coupled with the different vertical sensitivities and sources of error associated with
observations in different wavelength regions (e.g. Eskes and Boersma 2003; Ziemke et al., 2011; Miles et al.,
2015) means large-scale inconsistencies in time and space exist between the records of satellite
tropospheric column ozone (TCO$_3$) (as shown by Gaudel et al., 2018).
The work by Gaudel et al. (2018) was part of the Tropospheric Ozone Assessment Report (TOAR), which
represented a large global effort to understand spatio-temporal patterns and variability in TO$_3$. Their
investigation of ozonesondes (2003-2012) and products from nadir viewing satellites in polar orbits (three
from the Ozone Monitoring Instrument (OMI) (2005-2015/6) and two from the Infrared Atmospheric
Sounding Interferometer (IASI) (2008-2016)) displayed discrepancies in the spatial distribution, magnitude,
direction and significance of the TCO$_3$ trends. They noted that the records cover slightly different time
periods but were unable to provide any definitive reasons for these discrepancies beyond briefly suggesting
that differences in measurement techniques and retrieval methods were likely to be causing the observed
spatial inconsistencies. The range of potential definitions of the tropopause height used to derive TCO$_3$ from
these nadir-viewing profile products could also lead to differences between the satellite product absolute
values and their temporal evolution. While the 5 products discussed above use the same definition (i.e.
World Meteorological Organisation (WMO) 2 K/km lapse rate; WMO, 1957), several of the other products
analysed by Gaudel et al. (2018) did use other definitions.
The vertical sensitivity of each retrieved product (function of measurement technique and retrieval
methodology) used by Gaudel et al. (2018) will have had an impact on which part of the troposphere the O$_3$
signal is weighted towards. This is potentially one of the drivers behind the different OMI and IASI TCO$_3$
trends, where OMI showed predominantly positive trends between 60°S and 60°N while the opposite was

the case for IASI. The vertical sensitivity is represented by the "averaging kernel" (AK), which provides the relationship between perturbations at different levels in the retrieved and true profiles (Eskes and Boersma, 2003). Typically, for the products used by Gaudel et al., (2018), the peak AK sensitivities for $TO_3$ are in the 0-6 km range for OMI (Miles et al., 2015) and around 11-12 km for IASI (Keim et al., 2009), while there is a secondary peak at approximately 5 km (Boynard et al., (2009). In the case of the Rutherford Appleton Laboratory (RAL) Space OMI data, used in Gaudel et al., (2018), $TCO_3$ values were derived from retrieved surface – 450hPa layer average mixing ratios applied also to the overlying 450hpa – tropopause layer using ERA-Interim profiles. As the $TO_3$ values were derived from different (UV and IR) sensors and methodologies whose vertical sensitivities differ, they were likely representing $O_3$ controlled by different contributions of atmospheric processes (e.g. precursor emissions from the surface and stratosphere-troposphere exchanges). Therefore, $TCO_3$ trends from the different satellite products are not necessarily expected to be similar. The determination of the linear trend in a satellite $TCO_3$ record(s) can also be difficult as many factors (e.g. chemistry, emissions, deposition and transport) control ozone interannual variability, especially on time-periods of a decade or less (Barnes et al., 2016; Change et al., 2020; Fiore et al., 2022).

In this study, we undertake the first assessment of spatio-temporal variability in satellite-derived lower tropospheric column ozone ($LTCO_3$, surface-450 hPa) from three instruments over a consistent decade (2008-2017). In combination with an Earth System Model (ESM), we aim to quantify the impact of year-to-year spatiotemporal sampling, the satellite instrument uncertainties and the instrument vertical sensitivity on long-term $LTCO_3$ trends. We focus our analysis on North America, Europe and East Asia given their large emissions of ozone precursor gases and temporal variability. In our manuscript, **Section 2** discusses the satellite/ozonesonde datasets and model used, **Section 3** presents our results, and our discussion/ conclusions are summarised in **Sections 4 and 5**.

## 2. Methodology and Datasets
### 2.1. Satellite Datasets

The satellite products (see **Table 1**) used here are from nadir-viewing polar-orbiting platforms providing ozone sub-column profiles. This includes ozone profile data from the OMI product developed by the RAL Space and the IASI products from the Laboratoire d'aérologie (IASI-SOFRID) and the Université Libre de Bruxelles, in collaboration with the Laboratoire Atmosphères, Observations Spatiales (ULB-LATMOS) (IASI-FORLI). OMI and IASI are on NASA's Aura and Eumetsat's MetOp-A satellites in sun-synchronous low Earth orbits with local overpass times of 13.30 and 9.30, respectively. OMI and IASI are ultraviolet-visible (UV-Vis) and infrared (IR) sounders with spectral ranges of 270-500 nm (Boersma et al., 2008, Boersma et al., 2011) and 645-2760 $cm^{-1}$ (Illingworth et al., 2011), respectively. OMI has a spatial footprint at nadir of 24 km × 13 km, while IASI measures simultaneously in four fields of view (FOV, each circular at nadir with a diameter of 12 km) in a 50 km x 50 km square which are scanned across track to sample a 2200 km-wide swath (Clerbaux et al., 2009).

The OMI retrieval scheme is based on an optimal estimation (OE) approach, produced by RAL Space, which is described in detail by Miles et al., (2015). The retrieval schemes for IASI-FORLI and IASI-SOFRID $O_3$ are discussed in detail by Boynard et al., (2018) and Barret et al., (2020). The lowest sub-column in the OMI sub-column profile represents the surface-450 hPa layer (i.e. $LTCO_3$). For the IASI products, there were several sub-columns spanning the surface to 450 hPa range. Therefore, the IASI sub-columns were totalled up between the surface and the layer beneath or equal to the 450 hPa level. Where the 450 hPa level was located within a sub-column (i.e. was located between its bounding upper and lower pressure levels), the sub-column proportion between the lower pressure barrier and the 450 hPa level was determined and

added to the sub-columns below (i.e. towards the surface). For the ozone a priori profile, the RAL Space and
FORLI schemes use the ozone latitude vs month of year climatology of McPeters et al. (2007), while IASI-
SOFRID uses the dynamical ozone climatology described in Sofieva et al. (2014). However, the FORLI scheme
uses a single ozone profile (Boynard et al., 2018) derived from the McPeters et al. (2007) dataset, so has no
seasonality nor latitude dependence unlike the other retrieval schemes.
In this work, the OMI data were filtered for good quality retrievals where the geometric cloud fraction was
<0.2, the sub-column $O_3$ values were > 0.0, the solar zenith angle < 80.0°, the retrieval convergence flag = 1.0
and the normalised cost function was < 2.0. The IASI-FORLI data were filtered for a geometric cloud fraction
<0.13 (pre-filtered), degrees of freedom > 2.0, $O_3$ values > 0.0, solar zenith angle < 80.0° and the surface to
450 hPa sub-column $O_3$ / total column $O_3$ < 0.085. The IASI-SOFRID data were provided on a 1.0°×1.0°
horizontal grid (i.e. level 3 product, but at a daily temporal resolution – we use the daytime data in this
study) with filtering already applied in Barret et al., (2020). Here, only $O_3$ values > 0.0 were used. To remove
systematic biases between the satellite records, while maintaining the long-term inter-annual variability of
each record, ozonesondes were used to generate bias correction offsets (BCOs) (2008-2017) to help
harmonise the data sets (i.e. subtraction term in units of Dobson units, DU - as done in Russo et al. (2023)
and Pope et al. (2024)) and is discussed in the Supplementary Material (SM) (i.e. **S1**). By applying the BCOs,
this improves the robustness of the satellite datasets (in absolute terms). This is important when
intercomparing the products but also when using them to evaluate UKESM and determining the model's skill
to simulate $LTCO_3$ as used in this study (see **S4**).
Here, each ozonesonde profile was co-located with the nearest satellite retrieval within 500 km and 6 hours
to reduce spatiotemporal sampling biases (e.g. Keppens et al., 2019). The ozonesonde profile was then
interpolated in the vertical onto the satellite pressure grid where the sub-columns between pressure levels
were determined. The ozonesonde sub-column profiles were then convolved by the satellite averaging
kernels (AKs), which represent the satellite's sensitivity to retrieval ozone as a function of altitude. Thus,
allowing for a robust like-for-like comparison between the ozonesondes and the retrieved $LTCO_3$. The
application of AKs to ozonesonde profiles to evaluate satellite ozone products is discussed in detail by Pope
et al. (2023). The application of the AKs to the ozonesondes (and the model) is outlined in **Equation 1**:

$$sonde_{AK} = AK(sonde_{int} - apr) + apr \qquad (1)$$

where $sonde_{AK}$ is the modified ozonesonde sub-column profile (Dobson units, DU), $AK$ is the averaging kernel
matrix, $sonde_{int}$ is the sonde sub-column profile (DU) on the satellite pressure grid and $apr$ is the apriori
(DU). The application of the AKs to the ozonesondes is discussed in more detail in the SM **S1**.
To investigate long-term trends over North America, Europe and East Asia, the Hemispheric Transport of Air
Pollution (HTAP) regional sea-land mask (European Commission (2016); see **S2**, **Figure S5**), is used to sub-
sample the gridded satellite data for the respective regions and then generate average monthly time-series
for each product over each region of interest.  For the ozonesonde time-series for each HTAP region
investigated, only ozonesonde sites which are located within each HTAP region are selected. This results in
15, 13 and 6 ozonesonde sites for North America, Europe and East Asia, respectively. As ozonesonde data for
East Asia are all from Japan, Taiwan and Hong Kong, trends in ozone $LTCO_3$ will likely be different to
satellite/model trends over all East Asia.
In Section 3.2, where we discuss the impact of satellite retrieval errors on derived $LTCO_3$ linear trends, the
OMI and IASI-FORLI retrieval errors are provided in their product files but are not available for IASI-SOFRID.
Therefore, while not a perfect metric to represent the error in the IASI-SOFRID data, we use the standard
deviation in the monthly-spatial average of the regional time-series.


**2.2. United Kingdom Earth System Model (UKESM)**
The UK's Earth System Model, UKESM1.0, is a state-of-the-art ESM with fully interactive coupled component
models (e.g. atmosphere, ocean, land surface, atmospheric chemistry), which has been developed by the UK
Met Office and the Natural Environment Research Council (NERC). The detailed coupling of all the Earth
System components is described by Sellar et al. (2019). However, in this study, we run UKESM1.0 in an
atmosphere only configuration (e.g. similar to Archibald et al., (2020)). The aim is to use UKESM1.0 to
investigate long-term trends in $TO_3$ and help explore inconsistencies between satellite records, so it is
computationally more time efficient as only the atmospheric dynamics and chemistry components are
simulated. Over the 2008-2017 time period (with a 1-year spin up), the UKESM1.0 model tracers and
diagnostics (e.g. ozone, pressure) are output as 3D fields at sub-daily (6-hourly) time steps to allow robust
comparisons between the model and satellite data sets (i.e. model-satellite spatio-temporal co-location to
reduce representation biases and application of the satellite AKs to map the instrument vertical sensitivity
onto the model yielding like-for-like comparisons). The satellite AKs from OMI and IASI-FORLI are provided in
the level-2 files (i.e. an AK matrix per retrieval). However, the IASI-SOFRID AKs are provided from the gridded
level-3 data product (i.e. an AK matrix for each 1°×1° grid box).
Here, the UKESM1.0 land and atmosphere share a regular latitude–longitude grid with a resolution of 1.25°
×1.875° with 85 vertical levels on a terrain-following hybrid height coordinate with a model lid at 85 km
above sea level (50 levels are below 18 km). All the key inputs to the model from other Earth system
components (e.g. sea surface temperature (SST) and land surface vegetation) were prescribed from ancillary
files. The ocean and ice forcing are represented by the monthly Reynolds sea ice and SSTs data from the
National Oceanic and Atmospheric Administration (NOAA, https://climatedataguide.ucar.edu/climate-
data/). Solar forcings are provided by Phase 6 of the Coupled Model Intercomparison Project (CMIP6;
Matthes et al., 2017; Eyring et al., 2016), as is the stratospheric aerosol climatology to represent
contributions from volcanic eruptions (Sellar et al., 2019). The land cover is provided from output from the
land surface component of the ESM (JULES; Wiltshire et al., 2021) from a fully coupled historical simulation.
Anthropogenic and biomass burning emissions from Hoesly et al. (2018) and van Marle et al. (2017) are
prescribed for the period 2008 to 2014. After 2014, anthropogenic and biomass burning emissions are from
the Shared Socioeconomic Pathway (SSP, Rao et al., 2017) 2-4.5 (i.e. a middle-of-the-road climate and
emissions scenario).
Biological emissions are a climatology between 2001 and 2010 from the MEGAN-MACC data base
(Sindelarova et al., 2014), while natural emissions are from the Precursors of Ozone and their Effects in the
Troposphere (POET, http://accent.aero.jussieu.fr/database_table_inventories.php) based on 1990. Dry
deposition of $O_3$ to the land surface is represented by the Wesley scheme, which is applied as in O'Connor et
al., (2014). The model is also in a nudged or "specified dynamics" configuration (i.e. meteorological analyses
are used to "nudge" the model's meteorological variables, i.e. u- and v-wind components, and potential
temperature, towards reality; Telford et al., 2008) using 6-hourly reanalysis data from the European Centre
for Medium-Range Weather Forecasts (ECMWF) ERA-Interim product. A similar configuration of UKESM1.0
was used by Archibald et al., (2020), in which a thorough evaluation against multiple observations (e.g.
surface, aircraft and satellite) was carried out.

### 2.3. Trend Approach

LTCO$_3$ trends are calculated using the linear least squares fit approach of van der A et al., (2006; 2008), and
utilised by Pope et al., (2018) who investigated LTCO$_3$ trends. Here, the monthly LTCO$_3$ time-series are
represented by the function:

$$Y_t = C + BX_t + A\sin(\omega X_t + \phi) + N_t \qquad (2)$$

where $Y_t$ is the observed monthly LTCO$_3$ for month $t$, $X_t$ is the number of months since the start of the record,
$C$ is the first monthly mean LTCO$_3$ value of the record, $B$ is the monthly linear trend and $A\sin(\omega X_t + \phi)$ is the
seasonal model component (Weatherhead et al., 1998). $A$ is the amplitude, $\omega$ is the frequency (set to 1 year;
$\omega=\pi/6$) and $\phi$ is the phase shift. $C$, $B$, $A$ and $\phi$ are the fit parameters from the linear least squares fit. $N_t$
represents the model errors/residuals. The linear trend uncertainty, $\sigma_B$, represents the trend precision and is
calculated as:

$$\sigma_B = \left[ \frac{\sigma_N}{n^{\frac{3}{2}}} \sqrt{\frac{(1+\alpha)}{(1-\alpha)}} \right] \qquad (3)$$

where $n$ is the number of years, $\alpha$ is the autocorrelation in the residuals ($N_t$) and $\sigma_N$ is the standard deviation
in the residuals. As in van der A et al., (2006) and Pope et al., (2018), we calculate the autocorrelation for each
time-series using a lag of one-time step (i.e. one month). The autocorrelation in **Equation 2** is not accounted
for directly, so is factored into the trend uncertainty (**Equation 3**), as used and discussed by van der A et al.,
(2006) and Weatherhead et al., (1998), respectively.

## 3.  Results

A detailed evaluation of UKESM1.0 LTCO$_3$ through comparisons with the three satellite products and
ozonesondes is presented in **S4**. Overall, UKESM1.0 robustly simulates LTCO$_3$ spatially and seasonally in
comparison to the ozonesondes and satellite instruments (i.e. typically within the ozonesonde variability and
satellite uncertainty range).

### 3.1. UKESM1.0 and Satellite LTCO$_3$ Trends

#### 3.1.1.  North America

LTCO$_3$ trends from OMI, IASI-FORLI, IASI-SOFRID and ozonesondes are derived between 2008 and 2017 (i.e.
consistent time record for all instruments) using the linear-seasonal trend model (**Equation 2**). For each
satellite product, the corresponding UKESM1.0 time-series (with and without AKs) are analysed as well as
the satellite apriori. For the North America OMI metrics (**Figure 1 – top left, Table 2**), there is clear
seasonality in the apriori ranging between approximately 17.0 and 22.0 Dobson Units (DU). As this is based
on the climatology of McPeters et al., (2007), there is no trend and there is a very good model fit (i.e.
$R^2=1.0$). The key point is that, as a climatology, the apriori will have no trend but if there are substantial
temporal sampling differences between years, then an artificial trend could be introduced. OMI LTCO$_3$
ranges between 20.0 and 27.0 DU with substantial variability. There is a drop in LTCO$_3$ to 19.0 DU in 2009
before peaking at 25.0-27.0 DU between 2010 and 2015. Peak LTCO$_3$ then drops to 22.0-24.0 DU in 2016 and
2017. As a result, the linear-seasonal trend model, which does not account for interannual variations such as
this, only has a fit skill of $R^2=0.59$. The corresponding OMI LTCO$_3$ trend is -0.79 (-7.07, 5.48; 95% confidence
interval) DU/decade showing a negligible trend with a large uncertainty range. Here, -0.79 DU/decade is the
trend while the -7.07 and 5.48 DU/decade values are the 95% confidence interval. The UKESM1.0 LTCO$_3$
time-series ranges between 17.0 and 22.0 DU with clear seasonality, though somewhat less inter-annual
variation than OMI, and the linear-seasonal trend model therefore has a considerably better fit with $R^2$=0.95.
The model trend has the opposite sign at 0.21 (-0.37, 0.78) DU/decade. Here, the model trend is near-zero
with a relatively large uncertainty range (though not as sizable as OMI). When the AKs are applied to the
model, the trend switches sign to -0.57 (-1.58, 0.45) DU/decade and the linear-seasonal trend model fit
decreases in skill to $R^2$=0.90. The trend switch of sign, though small, is potentially linked to the application of
the AKs, which also increases LTCO$_3$ by 2.0-3.0 DU in general.
We also investigated the satellite degrees of freedom of signal (DOFS) over the lower troposphere (i.e.
surface to 450 hPa), which provides an estimate of the number of independent pieces of information in the
LTCO$_3$. The DOFS are calculated by taking the trace of the AK matrix over the lower tropospheric levels in the
satellite vertical grid. Overall, we found that the products for the three regions had negligible trends in their
time-series (i.e. within ±1.0 %/year) meaning that the information content of satellite LTCO$_3$ had remained
stable with time (see **S3**).
The IASI-FORLI LTCO$_3$ time-series (**Figure 1 – top right**) tends to be lower than OMI and ranges between 17.0
and 22.0 DU. There is a substantial negative IASI-FORLI trend (-1.42 (-2.35, -0.50) DU/decade; **Table 2**)
though as stated by Boynard et al., (2018) and Wespes et al., (2018), the input IASI Level-1 data sets into the
FORLI retrieval are not consistent with time; they suffer from a specific discontinuity in September 2010
which degrades the robustness of this trend. While we are aware of the artificial trend in the IASI-FORLI
dataset, it is still a valuable long-term product allowing us to quantify multiple factors (e.g. impact of AKs on
model tendencies/absolute values and year-to-year spatiotemporal sampling stability – i.e. near-zero trend
in the apriori). The apriori has a negligible trend but there is no clear seasonality in the apriori time-series. As
a result, the linear-seasonal trend model has a more limited fit skill (i.e. $R^2$=0.67). The impact of the satellite
AKs appears to have less impact for IASI-FORLI as both UKESM1.0 and UKESM1.0+AKs have time-series
ranging between approximately 17.0 and 21.0 (though slightly smaller UKESM1.0+AKs range) and linear-
seasonal trend model fits of $R^2$=0.93 and $R^2$=0.92, respectively. The corresponding trends are small at -0.13 (-
0.75, 0.49) and -0.32 (-0.82, 0.20) DU/decade, but the introduction of the AKs does move the UKESM1.0
trend slightly towards that of the satellite. Interestingly, while the application of the IASI-FORLI AKs to
UKESM marginally pushes the convolved model trend in LTCO$_3$ towards that of the satellite (which has a
substantial negative trend), the IASI-FORLI DOFS have small positive trends (0.37-0.57 %/year – see **S3**).
Therefore, there is minor scale, yet contrasting, discrepancy in how the vertical sensitivity is influencing the
long-term LTCO$_3$ trends.
For IASI-SOFRID (**Figure 1 – bottom left**), there is little difference between any of the time-series as they all
range between 16.0 and 21.0 DU with corresponding linear-seasonal trend model fits of $R^2$=0.94 to 0.98 and
negligible trends. The IASI-SOFRID and apriori trends are 0.12 (-0.59, 0.82; p = 0.74) and 0.11 (-0.17, 0.39)
DU/ decade; **Table 2)**, respectively, with the model showing near-zero trends in both cases. Given the close
agreement between the satellite and apriori time series and fit metrics, it is suggestive that IASI-SOFRID TO$_3$
is more closely confined to the apriori profile than are the other products.
The ozonesondes show a substantial trend of -1.15 (-2.0, -0.10) DU/decade, while the model trend sampled
as the sondes is -0.16 (-1.67, 1.35; p =0.63) DU/decade. The co-located model and ozonesonde linear-
seasonal trend model fits are $R^2$=0.62 and 0.64, respectively. The noise and lack of seasonality in the
ozonesonde time-series is slightly unexpected given the reasonable density of stations over North America,
though the spatial coverage and temporal sampling is much less than the satellite products.
**3.1.2. Europe**

In Europe, the OMI LTCO$_3$ values are larger than in North America, ranging between 19.0 and 30.0 DU (**Figure 2 – top left**). The same inter-annual variability exists, peaking between 2010 and 2015 with the minimum in 2009. Hence, the linear-seasonal trend model, which does not represent interannual variation, so has moderate skill and R$^2$=0.72. The corresponding trend is -0.80 (-7.29, 5.69) DU/decade, so has a similar direction and magnitude to that for North America, though is not substantial. The apriori ranges between 17.0 and 22.5 DU with a trend of -0.12 (-0.26, 0.03; **Table 2**) DU/decade. Given the relatively small trend and uncertainty range, unlike the OMI equivalent, it suggests there is unlikely to be an artificial trend arising through year-to-year spatiotemporal sampling changes in geographical sampling across the European region. UKESM1.0 LTCO$_3$ ranges between approximately 19.0 and 22.0 DU with a good linear-seasonal trend model fit of R$^2$=0.99 and a trend of -0.11 (-0.50, 0.29) DU/decade. As for North America, when the OMI AKs are applied, the UKESM LTCO$_3$ values systematically increase by 2.0-3.0 DU, move further away from the satellite apriori and more closely follow the variability of OMI (R$^2$ decreases slightly to 0.95). The trend tends towards that of OMI at -0.72 (-1.77, 0.32) DU/decade.

As in the case of North America, the European IASI-FORLI apriori has no seasonal cycle (and moderate R$^2$ of 0.48 in the linear-seasonal trend model fit) with a near-zero trend (0.09 (-0.09, 0.27) DU/decade) (**Figure 2 – top right**, **Table 2**). The IASI-FORLI data exhibit a substantial negative trend of -1.83 (-2.78, 0.89) DU/decade, again due to step changes in the IASI Level-1 processor, with a good linear-seasonal trend model fit of R$^2$=0.92. UKESM1.0 LTCO$_3$ trends, without and with AKs applied, are -0.28 (-0.77, 0.20) and -0.43 (-1.21, 0.35) DU/decade. Again, though a small change, the application of the AKs introduces a slight perturbation of the model trend compared to IASI-FORLI.

The IASI-SOFRID apriori, ranging between 17.0 and 21.0 DU, has a trend of 0.17 (-0.12, 0.45) DU/decade with good fit skill of R$^2$=0.98 (**Figure 2 – bottom left**). The IASI-SOFRID and UKESM1.0 metrics, with and without averaging kernels applied, are similar, with LTCO$_3$ trends of 0.05 (-0.91, 1.01;), -0.27 (-0.72, 0.19) and 0.08 (-0.33, 0.49) DU/decade, respectively, and with R$^2$ values between 0.93 and 0.98.

The ozonesonde monthly regional means (**Figure 2 – bottom right**) has a more pronounced time-series than North America, yielding a less noisy time-series of LTCO$_3$. Here, there is clear seasonality ranging between 17.0 and 24.0 DU with a large R$^2$ value of 0.95. The ozonesonde trend is relatively small at -0.61 (-1.39, 0.17) DU/decade while the UKESM1.0 equivalent is more substantial at -0.96 (-1.56, 0.35) DU/decade.

### 3.1.3. East Asia

For East Asia, OMI LTCO$_3$ again has both a pronounced seasonal cycle and inter-annual variability (19.0-27.0 DU), consistent with the other two regions discussed above (**Figure 3 – top left**, **Table 2**). This yields a moderate skill fit to the linear-seasonal trend model of R$^2$=0.52 and near-zero trend (-0.09 (-7.88, 7.70) DU/decade). The apriori has a trend of -0.25 (-0.71, 0.22) DU/decade, so year-to-year spatiotemporal sampling changes could be influencing the robustness of OMI retrieved time-series in this region. However, both the instrument and apriori trend uncertainties intersect with 0.0. UKESM1.0 LTCO$_3$ ranges between approximately 16.0 and 22.0 DU with a good fit R$^2$ of 0.98. Like the other regions, the application of the OMI AKs increases the model values systematically by several DUs. The UKESM1.0 LTCO$_3$ trend is -0.16 (-0.94, 0.62) DU/decade, which is small, but the AKs increase the trend magnitude to -0.62 (-2.24, 1.00) DU/decade, which moves it away from the OMI trend.

IASI-FORLI (**Figure 3 – top right**, **Table 2**), like the other two regions, has a substantial negative trend of -1.52 (-2.16, 0.88) DU/decade. The apriori again exhibits virtually no seasonal cycle (low fit skill of R$^2$=0.21) and negligible year-to-year spatiotemporal sampling differences yielding a near-zero trend of -0.03 (-0.22, 0.16)

DU/decade. For UKESM1.0, the East Asian seasonal range is much larger than other regions, ranging
between 17.0 and 27.0 DU (i.e. seasonal amplitude of approximately ±5.0 DU). When the AKs are applied,
this range shrinks to approximately 19.0 to 23.0 DU, more in-line with the IASI-FORLI $LTCO_3$ values. The
corresponding model trends are -0.03 (-0.62, 0.56) DU/decade and -0.29 (-0.80, 0.22) DU/decade, so the AKs
are pushing the model tendency towards that of the instrument, though the impact is small in absolute
terms (large in relative terms).
IASI-SOFRID and its apriori $LTCO_3$ seasonality are again very similar, ranging between 16.0 and 21.0 DU with
very little interannual variability and with linear seasonal trend model fit skills of $R^2$=0.96 and 0.98 (**Figure 3 –**
**bottom left**, **Table 2**). The IASI-SOFRID and apriori linear trends are therefore also consistent at -0.19 (-1.01,
0.63) and -0.15 (-0.73, 0.58) DU/decade. The UKESM1.0 seasonal variability is again large, between 17.0 and
26.0 DU, and, as in the case of IASI-FORLI, when the instrument AKs are applied to the model, the seasonal
range shrinks (i.e. 16.0-22.0 DU) to be much closer to those of the retrieval and its prior. The model trends
are -0.42 (-0.97, 0.13) and -0.24 (-0.67, 0.20) (with AKs) DU/decade, where there is a minor shift in the model
tendency towards that of IASI-SOFRID and its prior.
For the ozonesondes (**Figure 3 – bottom right**), there is a substantial $LTCO_3$ trend of 3.17 (0.16, 6.17)
DU/decade with a fit skill of $R^2$=0.79, which is larger than those for North America and Europe. $LTCO_3$
increases from 18.0-25.0 in 2008 to 21.0-28.0 in 2011. This remains similar in 2012 and 2013 before
dropping by several DUs between 2014 and 2017. The UKESM1.0 sampled as the ozonesondes has
considerably less inter-annual variability with a smaller trend of 0.37 (-0.90, 1.64) DU/decade. Therefore,
UKESM1.0 and the satellite product trends are generally smaller (in magnitude) than the ozonesonde
tendencies. However, it is worth considering that there are only a few sites (e.g. Hong Kong and Taiwan)
where ozonesonde data is available in East Asia.
**3.2. Influence of Satellite Averaging Kernels on UKESM1.0 $LTCO_3$**
To investigate the impact of applying the satellite averaging kernels to UKESM1.0, and thus learn something
about vertical sensitivity influence on retrieved $LTCO_3$, three different metrics are considered for the 2008 to
2017 time-period. These are the absolute $LTCO_3$ value, amplitude of the $LTCO_3$ seasonal cycle and the linear
trend. These metrics are compared for the satellite, the satellite ± error term, the apriori, UKESM1.0 and
UKESM1.0+AKs for the three regions discussed above.
From **Figure 4**, average OMI $LTCO_3$ is approximately 22.0, 24.0 and 23.0 DU for North America, Europe and
East Asia, respectively. This represents a substantial deviation away from the apriori values of 17.5, 20.0 and
16.0 DU, respectively. However, the average error term for OMI $LTCO_3$ is sizeable at approximately ±8.0 to
±9.0 DU for all regions. The average UKESM1.0 value for each region is approximately 19.5, 21.5 and 19.0 DU
but the application of the AKs increases this by several DU to 22.0, 24.0 and 21.0 DU. In comparison, mean
values for both IASI products vary less between the three geographical areas: IASI-FORI (IASI-SOFRID) $LTCO_3$
values are 20.0 (18.5), 19.0 (18.5) and 22.0 (18.0) DU, respectively. The corresponding error ranges, in
comparison with OMI, are smaller between 17.0 and 23.0 (16.0 and 21.5), 16.0 and 21.5 (16.0 and 21.0) and
18.0 and 23.5 (14.5 and 21.5) DU for North America, Europe and East Asia, respectively. With the IASI-FORLI
AKs applied to UKESM1.0, $LTCO_3$ decreases from 19.5 to 19.25 DU, 21.25 to 19.5 DU and 22.75 to 21.25 DU
for the three regions. For IASI-SOFRID, there is a decrease from 21.0 to 19.5 DU in Europe and a decrease
from 22.0 to 19.5 DU in East Asia, while no change occurs in North America. Overall, OMI has the largest
error range and the application of the AKs to UKESM1.0 systematically increases the model $LTCO_3$ time-
series by several DU. The opposite occurs for the IASI products where there is a smaller decrease to
UKESM1.0 $LTCO_3$ of 1.0-2.0 DU. The error ranges are also smaller than that of OMI.
In terms of the LTCO$_3$ seasonal amplitude (**Figure 5**), OMI (including the error terms) is approximately 2.6
(for all) DU, 3.3-3.8 DU and 2.3-2.6 DU for North America, Europe and East Asia. The apriori seasonal
amplitude ranges from 2.7 to 2.9 DU across the regions. The IASI-FORLI averages (including the error terms)
tend to be lower than OMI but have similar seasonal ranges. North America, Europe and East Asia have
amplitudes of 2.3-2.5 DU, 2.3-2.5 DU and 1.6-1.8 DU, respectively. It is noteworthy that this seasonal cycle is
despite the IASI-FORLI prior exhibiting virtually no seasonal cycle at all. IASI-SOFRID has a European range of
2.4-2.6 DU, and comparable ranges for North America and East Asia at 1.8-2.5 DU and 2.3-3.0 DU. Therefore,
seasonal amplitude in IASI-SOFRID is more sensitive to the error metric but as the "error" term is based on
the LTCO$_3$ standard deviation, given the lack of an error term in the product, it is unsurprising that there is
more variability in the seasonal amplitude. For the OMI comparisons, the application of the AKs to
UKESM1.0 shifts the simulated amplitude slightly upwards from 2.0 to 2.1 DU, 3.1 to 3.3 DU and 4.0 to 4.4
DU for the respective regions. The IASI-FORLI AK impacts are a decrease from 1.9 to 1.4 DU, 3.0 to 2.1 DU
and 4.2 to 1.9. For IASI-SOFRID, the corresponding impact on UKESM1.0 is 2.2 to 2.4 DU, 3.3 to 2.9 and 4.5 to
3.2 DU. Therefore, the OMI AKs have a minimal impact, increasing the model seasonal amplitude by 0.1-0.3
DU, but the IASI products suppress the simulated amplitude by 1.0-2.0 DU at the most extreme.
The impact of the satellite LCTO$_3$ error terms on the derived linear trends are shown in **Figure 6**. For OMI,
the range in trends calculated (i.e. satellite ± error term) is approximately -1.50 (-7.04, 4.04) to -0.09 (-6.98,
6.81) DU/decade, -1.65 (-6.92, 3.62) to 0.05 (-7.44, 7.53) DU/decade and -1.05 (-6.61, 4.52) to 0.87 (-8.24,
9.98) DU/decade for North America, Europe and East Asian, respectively. The IASI-FORLI trends (i.e. satellite
± error term) are substantial ranging from -1.50 (-2.51, -0.50) to -1.34 (-2.21, -0.47) DU/ decade, -1.87 (-2.87,
-0.87) to -1.80 (-2.72, -0.88) DU/decade and -1.62 (-2.27, -0.98) to -1.42 (-2.06, -0.78) for the three regions,
respectively. The corresponding IASI-SOFRID trends were 0.09 (-0.48, 0.66) to 0.14 (-0.59, 0.88) DU/decade, -
0.07 (-0.91, 0.78) to 0.16 (-0.74, 1.07) DU/decade and -0.30 (-1.02, 0.42) to -0.08 (-0.73, 0.58) DU/decade,
respectively. Therefore, only the IASI-FORLI trends (i.e. satellite ± error term) are substantially different from
zero (i.e. $p < 0.05$). However, that is due in part to discontinuities in the input meteorological data used to
generate this version of the product (Boynard et al., 2018).
The application of the OMI AKs to UKESM1.0 had the largest impacts on the simulated trends with changes
in a negative direction from of 0.21 (-0.37, 0.78) to -0.57 (-1.58, 0.45) DU/decade, -0.11 (-0.50, 0.29) to -0.72
(-1.77, 0.32) DU/decade and -0.16 (-0.94, 0.62) to -0.62 (-2.24, 1.00) DU/decade for the respective regions.
IASI-FORLI AKs introduced small decreases from -0.13 (-0.75, 0.49) to -0.32 (-0.82, 0.20) DU/decade, -0.28 (-
0.77, 0.20) to -0.43 (-1.21, 0.35) DU/decade and -0.03 (-0.62, 0.56) to -0.29 (-0.80, 0.22) DU/decade. IASI-
SOFRID AKs introduced small increases in the LTCO$_3$ trend from -0.24 (-0.85, 0.37) to -0.04 (-0.53, 0.45)
DU/decade, -0.27 (-0.72, 0.19) to 0.08 (-0.33, 0.49) DU/decade and -0.42 (-0.97, 0.13) to -0.24 (-0.67, 0.20)
DU/decade.
As the absolute model trends are small, it is difficult to determine the impact of the AKs on the simulated
trends. In relative terms, it can have impacts of several 100% but the model and model+AK trend ranges
(95% confidence interval) always intersect. Therefore, in an attempt to derive more substantial UKESM1.0
LTCO$_3$ trends (without and with AKs applied), to assess the maximum impact the AKs can have on UKESM
LTCO$_3$ trends, the modelled data were sorted from lowest to highest and the trend re-calculated. In North
America, this approach forced positive model trends, sub-sampled to OMI, IASI-FORLI and IASI-SOFRID, of
0.73 (0.22, 1.25), 0.64 (-3.50, 4.77) and 0.80 (0.41, 1.19) DU/decade. When the AKs were applied, it yielded
trends of -0.74 (-1.89, 0.40), 0.55 (0.08, 1.03) and 0.58 (0.24, 0.92) DU/decade. In Europe, this forced positive
trends model trends, of 0.62 (0.14, 1.10), 0.37 (-0.05, 0.79) and 0.46 (0.09, 0.84) DU/decade, respectively.
With the AKs applied, the trends become 0.47 (-0.51, 1.44), 0.28 (-0.38, 0.94) and 0.10 (-0.32, 0.51)
DU/decade. Finally, in East Asia, the forced model trends are 0.90 (0.34, 1.47), 0.66 (0.15, 1.17) and 0.63
(0.26, 1.00) DU/decade. The application of the AKs introduced model trends of 1.02 (-0.04, 2.09), 0.08 (-0.44,
0.61) and 0.20 (-0.20, 0.61) DU/decade.
Even with forced trends in the UKESM1.0 regional time-series, the trends are relatively small (i.e. typically
less than 1.0 DU/decade in magnitude). Therefore, the application of the AKs to the forced UKESM LTCO$_3$
time-series still yields small scale changes in tendencies and there is overlap in the two model trend
uncertainty ranges (i.e. 95% confidence level). However, in relative terms, the trend changes are larger (e.g.
>100% in multiple cases) and there is often a shift of the modelled LTCO$_3$ trend uncertainty range either
intersecting or no longer intersecting with zero (i.e. a shift in p-value regime from <0.05 to >0.05). Therefore,
in modelled and satellite datasets with more substantial trends, the impacts of the AKs, and thus the satellite
vertical sensitivity, on LTCO$_3$ trends would be much greater and potentially help pinpoint sources of
differences between satellite products in their TO$_3$ temporal evolution.
**3.3. Diurnal Variability on Regional LTCO$_3$ and Temporal Evolution**
As TO$_3$ varies diurnally due to meteorological and photochemical processes (e.g. Gaudel et., 2018), the
different satellite overpass times (i.e. Aura and MetOp-A daytime overpasses are around 13:30 and 09:30
local time, respectively) will likely influence the spatial distributions of TO$_3$ which OMI and IASI will retrieve.
In principle, this could therefore explain some differences between the two sensors and their long-term
LTCO$_3$ trends. Here, the model is a useful tool to help investigate this and we used the 6-hourly output to
derived the UKESM simulated LTCO$_3$ spatial distributions at the Aura (13.30 LT) and MetOp-A (09.30 LT) day-
time overpasses. These model fields were then used to calculate regional time-series for North America,
Europe and East Asia. For each region and month, between 2008 and 2017, we calculated the regional
average absolute difference (i.e. from the selection of model grid cells which fell within the HTAP-2 mask for
a specific month) and the standard deviation of the absolute differences between the overpass times. Here,
across all months and regions, we found the peak average absolute difference (13:30 LT – 09:30 LT) and
standard deviation to be small at 2.03 and 2.56%, respectively. For the long-term trends, across all regions
and overpass times, all of the UKESM trends were smaller than ±0.5 DU/decade. Therefore, the model LTCO$_3$
regional trends are negligibly different between overpass times. This might not be surprising given the
negligible model trends in the satellite spatio-temporal trend comparisons (see **Section 3.1**), but the actual
absolute differences (average and range) in simulated LTCO$_3$ are also small supporting the argument that on
the regional scale, the day-time diurnal cycle differences between satellite overpass times has limited
influence on the reported satellite trend discrepancies (e.g. in Gaudel et al., 2018).
**4.  Discussion**
Investigation of satellite LTCO$_3$ focussed on 2008 to 2017, representing a decade of overlap of the OMI and
IASI records. The analysis focussed on North America, Europe and East Asia as these regions are subject to
large emissions of and temporal changes in O$_3$ precursor gases. LTCO$_3$ is typically spatially homogeneous
with shallow gradients between background and source-induced O$_3$ concentrations. Secondly, individual
retrievals of LTCO$_3$ are often associated with large uncertainties (e.g. random and systematic uncertainties).
There are multiple contributory factors concerning both instrumental attributes (notably spectroradiometric
noise and calibration accuracy) and variability in geophysical variables which influence radiative transfer and
vertical sensitivity (e.g. stratospheric ozone, cloud and aerosol, water vapour, surface spectral
reflectivity/emissivity and pressure and temperature profile) which can result in LTCO$_3$ time-series with
substantial variability/noise when derived at high spatial resolution (e.g. when deriving time-series from data
gridded at 0.5° or 1.0°). Therefore, we undertake our analysis at the regional (e.g. continental) scale where
more satellite retrievals are included in time-series monthly means yielding a reduction in the random error
component of the sample.
Ideally, this analysis would have utilised several more records (e.g. several UV-Vis and IR products) to
quantify long-term trends in $LTCO_3$ and investigate the potential reasons for any discrepancies, as shown by
Gaudel et al., (2018) for $TCO_3$. While RAL Space, and other providers, have generated UV-Vis profile $O_3$
products for more instruments, e.g. from the Global Ozone Monitoring Experiment 1 & 2 (GOME-1 & GOME-
2) and the SCanning Imaging Absorption spectroMeter for Atmospheric CartograpHY (SCIAMACHY), the
GOME-1 and SCIAMACHY records do not overlap for as long with IASI and step changes in the GOME-2A
Level-1 processing scheme used to produce the available $LTCO_3$ Level-2 version mean it is not sufficiently
homogeneous (see Pope et al., (2023)). For the IR instruments, other potential sensors include the
Tropospheric Emissions Spectrometer (TES; Richards et al., 2008) and the RAL Space IASI Extended Infrared
Microwave Sounding (IMS; Pimlott et al., 2022) scheme applied to IASI. Unfortunately, the TES record only
covers 2005 to 2013, with decreasing spatial coverage with time, and at the time of this work the IASI-IMS
product had only been processed on a sub-sampled basis of 1 in 10 days.
In this work, we some find discrepancies in the observed long-term tendencies from the utilised $LTCO_3$
products in these northern hemispheric regions. The OMI product is subject to large-scale interannual
variability over the 2008-17 decade, in comparison with which the underlying linear trends are small in
absolute terms with large confidence ranges (i.e. 95% confidence intervals) intersecting with zero. However,
the OMI $LTCO_3$ product has been shown to be stable over this period relative to ozonesondes by Pope at el.,
(2023). IASI-FORLI has substantial negative $LTCO_3$ tendencies, but this is driven by a specific discontinuity in
2010 due to inhomogeneity in Eumetsat (water vapour, temperature) data used in IASI-FORLI Level-2
processing (Boynard et al., 2018; Wespes et al., 2018). It induces an artificial drift that explains the
substantial negative $LTCO_3$ trends reported here and in Gaudel et al., (2018). The IASI-SOFRID $LTCO_3$ and
apriori are very similar, with little inter-annual variability, which suggests that the IASI-SOFRID $O_3$ retrieval in
this height-range is more constrained by the apriori (i.e. less $TO_3$ sensitivity than the other products – see
**S3**). Importantly, analysis of the three products' apriori $LTCO_3$ records show negligible trends meaning that
year-to-year spatiotemporal sampling differences (i.e. the number of retrievals used in the spatial-monthly
regional averages) are not skewing long-term satellite trends. In summary: any underlying linear trend in
$LTCO_3$ occurring during the decade 2008-17 was masked by interannual variability in the OMI retrieval and
by constraint to the apriori in the IASI-SOFRID retrieval and, although substantial for IASI-FORLI retrieval,
that is due to changing meteorological inputs to the data processing (Boynard et al., 2018; Wespes et al.,
492   2018).

For UKESM1.0, the model exhibits negligible temporal variability in $LTCO_3$ for all regions and instruments'
samplings. Modelled $LTCO_3$ trends never exceeded 1.0 DU/decade in magnitude, all of which were deemed
to be insignificant due to large associated p-values by the linear-seasonal trend model detailed in **Section 2.3**
and **Equations 2** & **3**. The ozonesondes for each region were included to ground truth the model and satellite
trends. The North American sites' $LTCO_3$ time-series was relatively noisy and exhibited considerable inter-
annual variability in its seasonal cycle. The comparatively low level of inter-annual variability in the European
UKESM1.0 record of $LTCO_3$ was in good agreement with the ozonesondes, and so was its low trend,
providing confidence in the model over that region. For East Asia, the interannual variability differed
substantially between UKESM1.0 and ozonesondes and the reported ozonesonde trend was significantly
much larger than for UKESM1.0. Therefore, when considering UKESM1.0 and the ozonesondes, no consistent
$LTCO_3$ trends can be determined for any of the regions. Overall, taking all data sets into account, $LTO_3$

appears to have neither increased nor decreased markedly over these three regions between the beginning and end of the study decade (i.e. 2008 to 2017).

One key aspect of this work was to exploit UKESM1.0 to determine the importance of vertical sensitivity on retrieved $LTO_3$ and how this influences the reported long-term tendency. In terms of the absolute model trends (with and without the satellite AKs), the impact on $LTCO_3$ was small with typically near-zero tendencies and large uncertainty ranges (i.e. the 95% confidence interval). In relative terms, the changes in model trend values were more substantial in the order of 100%. To explore this further, the UKESM1.0 $LTCO_3$ time-series (with and without the satellite AKs) were sorted from lowest to highest (based on annual averages) to impose the most substantial trend in the model data. When the trends were re-calculated, the largest model $LTCO_3$ trends ranged between 0.37 and 0.90 DU/decade. When the AKs were applied, the $LTCO_3$ trends ranged from -0.74 to 1.02 DU/decade. Again, in relative terms, this represents a large impact of the AKs on simulated $LTO_3$ tendencies but in absolute terms, these are small changes. Though, it should be noted that many of the 95% confidence intervals for these trends either shifted to intersect with zero or vice versa once the AKs were applied to the model. Gaudel et al., (2018) suggested two potential reasons for the $TCO_3$ trend discrepancies in their study:

- Time varying instrument biases/drift.
- The impact of satellite vertical sensitivity.

A further two important reasons are:

- Changes over time in latitude/longitude domains sampled by satellite sensors (e.g. GOME-1 has substantial issues after 2003).
- The time-period used for the trend analysis.

As stated by Boynard et al., (2018) and Wespes et al., (2018), the IASI-FORLI-v20151001 product has an artificial negative drift with time explained by a discontinuity found in the Level-2 meteorological inputs taken from Eumetsat. However, in the near future, a new consistent IASI-FORLI ozone climate data record will be available using homogeneous Level-1 and Level-2 Eumetsat meteorological data. Analysis of OMI $LTCO_3$ by Pope et al., 2023 showed OMI $LTCO_3$ to be temporally stable against ozonesondes. A similar analysis (not shown here) indicates IASI-SOFRID $LTCO_3$ to also be temporally stable with near-zero drift in bias. For the satellite vertical sensitivity, some of our results were unexpected. While the application of the AKs to UKESM1.0 can substantially shift the simulated absolute $LTO_3$ values and squash/stretch the seasonal amplitude, the impact on the simulation $LTCO_3$ tendencies are small in absolute terms. In relative terms, the impacts can be large (e.g. 100% change in trend rate). However, as the UKESM1.0 simulated $LTCO_3$ trends are generally near-zero, it is difficult to confidently say either way if the vertical sensitivity, when retrieving $LTCO_3$, is important for influencing long-term tendencies, even when a more substantial trend is forced upon UKESM1.0. Future work on this would probably need to look at artificial model data which already has substantial $TO_3$ trends in it (e.g. 5.0 or 10.0 DU/decade). This will obviously not match reality but would provide some further quantification on how important vertical sensitivity is from different instruments/sounders in $LTO_3$ trend determination.

As for year-to-year spatiotemporal sampling, our results suggest negligible trends for the product $LTCO_3$ apriori time-series and thus monthly sampling biases are unlikely to be introducing artificial trends as the apriori datasets are trendless. Finally, the time-period over which the trend analysis is undertaken is critically important. Gaudel et al., (2018), using the available data at the time, focussed on 2005-2015/6 and 2008-2015/6 for the OMI and IASI products they used. For the IASI products, using a slightly extended time-period,

the trends show similar tendencies. However, for OMI, 2016 and 2017 represent lower years of $TO_3$. As a
result, this dampens the strong significant positive trends reported by Gaudel et al., (2018) in $TCO_3$. It is
notable that the substantial positive increase in tropical $LTO_3$ between 1995 and 2017 reported by Pope et
al., (2023) from a series of UV-Vis sounders, included the same OMI global dataset as that is used here,
further suggests the selection of time period and geographical region to be crucial in regard to the role of
interannual variability on linear trend detection.

## 5. Conclusions

Gaudel et al., (2018) undertook a multi-satellite analysis of long-term trends in tropospheric column ozone
($TCO_3$). They found large scale differences between these products with no clear consensus on the signs or
drivers of these $TCO_3$ trends. To avoid complications with tropopause definition and reduce influence of
stratospheric ozone on retrieved values, this study has undertaken a detailed follow-up assessment of
decadal trends in $LTCO_3$ (surface – 450 hPa layer) rather than $TCO_3$ exploiting ozonesonde records, model
simulations and accounting carefully for satellite $O_3$ metrics (e.g. averaging kernels, AKs, apriori information
and satellite uncertainties). We have focussed on $LTCO_3$ data sets from Ozone Monitoring Instrument (OMI)
produced by the RAL Space scheme and from Infrared Atmospheric Sounding Interferometer produced by
the IASI-FORLI and IASI-SOFRID schemes, for which there were consistent records from 2008-2017.
Evaluation of satellite $LTO_3$ from these three products over the North American, European and East Asian
regions resulted in linear trends which varied over a small range close to zero and with confidence intervals
intersecting with zero. This was consistent with simulations from the UK Earth System Model (UKESM1.0).
There were no large-scale trends in the apriori information, so changes in satellite year-to-year
spatiotemporal sampling has not been driving inconsistencies between products. When convolving
UKESM1.0 with the satellite AKs (i.e. to assess the impact of the satellite vertical sensitivity) it did change the
size of the model trend, and in some instances, the direction of the trend, but as the simulated $LTO_3$ trends
were small and insignificant, they had limited influence. Overall, our results show that changes in $LTO_3$
during the decade 2008-2017 in North America, Europe and East Asia were dominated by variability in
processes which control $LTO_3$ on shorter timescales.
In the near future, the new European polar orbiting mission MetOp Second Generation will include IASI Next
Generation and Sentinel-5 UV/VIS sounders to provide height-resolved ozone products to extend current
missions through to the mid-2040s. This will be supplemented by the new USA Near Earth Orbit Network
(NEON) series as a replacement for the Joint Polar Satellite System (JPSS). The Geostationary Environment
Monitoring Spectrometer (GEMS) and Tropospheric Emissions: Monitoring of Pollution (TEMPO) have also
recently been launched and there will be new geostationary platforms: the Infrared Sounder (IRS) and
Sentinel-4 UV/VIS sounder on Europe's Meteosat-Third Generation (MTG-S), again through to the mid-
2040s, and the USA Geostationary Extended Observations (GeoXO) series. Overall, these platforms will
provide large volumes of data (e.g. diurnal observations) and over a long-time scale on tropospheric ozone
for future regional trend analyses.

**Acknowledgements**

This work was funded by the UK Natural Environment Research Council (NERC) by providing funding for the
National Centre for Earth Observation (NCEO, award reference NE/R016518/1), the NERC funded UKESM
Earth system modelling project (award reference NE/N017978/1) and funding from the European Space
Agency (ESA) Climate Change Initiative (CCI) post-doctoral fellowship scheme (award reference
4000137140). For UKESM1.0 model runs, we acknowledge use of the Monsoon2 system, a collaborative
facility supplied under the Joint Weather and Climate Research Programme, a strategic partnership between
the Met Office and NERC. IASI is a joint mission of EUMETSAT and the Centre National d'Etudes Spatiales
(CNES, France). The IASI-SOFRID research was conducted at LAERO with some financial support from the
CNES French spatial agency (TOSCA–IASI project). The authors acknowledge the AERIS data infrastructure for
providing access to the IASI-FORLI data, ULB-LATMOS for the development of the FORLI retrieval algorithm,
and the AC SAF project of the EUMETSAT for providing IASI-FORLI data used in this paper. Anna Maria
Trofaier (ESA Climate Office) provided support and advice throughout the fellowship.

**Data Availability**

The IASI-FORLI and IASI-SOFRID data can be obtained from https://iasi.aeris-data.fr/O3 and https://iasi-
sofrid.sedoo.fr/. The RAL OMI data is available via the NERC Centre for Environmental Data Analysis (CEDA)
Jasmin platform subject to data requests. However, the RAL Space satellite data, as well as the UKESM1.0
simulations, will be uploaded to the Zenodo open access portal (https://zenodo.org/) if this manuscript is
accepted for publication in ACP after the peer-review process. The ozonesonde data for WOUDC, SHADOZ
and NOAA is available from https://woudc.org/, https://tropo.gsfc.nasa.gov/shadoz/ and
https://gml.noaa.gov/ozwv/ozsondes/.

**Author Contributions**

RJP conceptualised, planned and undertook the research study. BB, ELF, BJK, RS, BGL, AB and CW provided
the OMI and IASI ozone data and advice on using the products and their analysis. FO and MD provided
advice and expertise on using and running UKESM. CR provided advice and help during RP's ESA CCI
fellowship. Scientific and technical contributions came from MPC, WF, MAP, SSD and RR. RJP prepared the
manuscript with input from all co-authors.

**Conflicts of Interest**

The authors declare no conflicts of interest.

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

**Figures & Tables:**

| Data Provider | Satellite Profile Products & Version | Product Link | Data Range | Data Size |
|---|---|---|---|---|
| RAL Space | OMI–fv214 | http://www.ceda.ac.uk/ | 2004-2018 | 1442 GB |
| ATMOS-ULB | IAS-FORLI-v20151001 | https://iasi.aeris-data.fr/catalog/ | 2008-2019 | 9.1 TB |
| Université de Toulouse | IASI-SOFRID vn3.5 | https://iasi-sofrid.sedoo.fr/ | 2008-2017 | 3.0 TB |

***Table 1:*** *List of the satellite ozone profile data sets.*

| Satellite | Quantity | Trend | Trend Lower | Trend Upper | p-value | Fit ($R^2$) |
|---|---|---|---|---|---|---|
| OMI – North America | Trend | -0.79 | -7.07 | 5.48 | 0.80 | 0.58 |
| | Trend Error 1 | -1.50 | -7.04 | 4.04 | 0.59 | 0.68 |
| | Trend Error 2 | -0.09 | -6.98 | 6.81 | 0.98 | 0.50 |
| | Apriori Trend | -0.05 | -0.21 | 0.11 | 0.56 | 1.00 |
| | UKESM Trend | 0.21 | -0.37 | 0.78 | 0.47 | 0.95 |
| | UKESM+AKs Trend | -0.57 | -1.58 | 0.45 | 0.26 | 0.90 |
| | UKESM Trend Forced | 0.73 | 0.22 | 1.25 | 0.00 | 0.95 |
| | UKESM+AKs Trend Forced | -0.74 | -1.89 | 0.40 | 0.20 | 0.89 |
| FORLI – North America | Trend | -1.42 | -2.35 | -0.50 | 0.00 | 0.93 |
| | Trend Error 1 | -1.34 | -2.21 | -0.47 | 0.00 | 0.93 |
| | Trend Error 2 | -1.50 | -2.51 | -0.50 | 0.00 | 0.93 |
| | Apriori Trend | 0.00 | -0.11 | 0.12 | 0.94 | 0.67 |
| | UKESM Trend | -0.13 | -0.75 | 0.49 | 0.67 | 0.93 |
| | UKESM+AKs Trend | -0.32 | -0.83 | 0.20 | 0.22 | 0.92 |
| | UKESM Trend Forced | 0.64 | -3.50 | 4.77 | 0.76 | 0.46 |
| | UKESM+AKs Trend Forced | 0.55 | 0.08 | 1.03 | 0.02 | 0.93 |
| SOFRID – North America | Trend | 0.12 | -0.59 | 0.82 | 0.74 | 0.94 |
| | Trend Error 1 | 0.14 | -0.59 | 0.88 | 0.70 | 0.90 |
| | Trend Error 2 | 0.09 | -0.48 | 0.66 | 0.75 | 0.94 |
| | Apriori Trend | 0.11 | -0.17 | 0.39 | 0.43 | 0.98 |
| | UKESM Trend | -0.24 | -0.85 | 0.37 | 0.44 | 0.95 |

| | | | | | | |
|---|---|---|---|---|---|---|
| | UKESM+AKs Trend | -0.04 | -0.53 | 0.45 | 0.87 | 0.97 |
| | UKESM Trend Forced | 0.80 | 0.41 | 1.19 | 0.00 | 0.97 |
| | UKESM+AKs Trend Forced | 0.58 | 0.24 | 0.92 | 0.00 | 0.98 |
| OMI -Europe | Trend | -0.80 | -7.29 | 5.69 | 0.80 | 0.71 |
| | Trend Error 1 | -1.65 | -6.92 | 3.62 | 0.53 | 0.76 |
| | Trend Error 2 | 0.05 | -7.44 | 7.53 | 0.99 | 0.67 |
| | Apriori Trend | -0.12 | -0.26 | 0.03 | 0.10 | 1.00 |
| | UKESM Trend | -0.11 | -0.50 | 0.29 | 0.59 | 0.99 |
| | UKESM+AKs Trend | -0.72 | -1.77 | 0.32 | 0.16 | 0.95 |
| | UKESM Trend Forced | 0.62 | 0.14 | 1.10 | 0.01 | 0.98 |
| | UKESM+AKs Trend Forced | 0.47 | -0.51 | 1.44 | 0.34 | 0.94 |
| FORLI - Europe | Trend | -1.83 | -2.78 | -0.89 | 0.00 | 0.92 |
| | Trend Error 1 | -1.80 | -2.72 | -0.88 | 0.00 | 0.93 |
| | Trend Error 2 | -1.87 | -2.87 | -0.87 | 0.00 | 0.92 |
| | Apriori Trend | 0.09 | -0.09 | 0.27 | 0.32 | 0.48 |
| | UKESM Trend | -0.28 | -0.77 | 0.20 | 0.25 | 0.98 |
| | UKESM+AKs Trend | -0.43 | -1.21 | 0.35 | 0.27 | 0.94 |
| | UKESM Trend Forced | 0.37 | -0.05 | 0.79 | 0.08 | 0.98 |
| | UKESM+AKs Trend Forced | 0.28 | -0.38 | 0.94 | 0.40 | 0.93 |
| SOFRID - Europe | Trend | 0.05 | -0.91 | 1.01 | 0.92 | 0.93 |
| | Trend Error 1 | 0.16 | -0.74 | 1.07 | 0.72 | 0.91 |
| | Trend Error 2 | -0.07 | -0.91 | 0.78 | 0.87 | 0.93 |
| | Apriori Trend | 0.17 | -0.12 | 0.45 | 0.24 | 0.98 |
| | UKESM Trend | -0.27 | -0.72 | 0.19 | 0.24 | 0.98 |
| | UKESM+AKs Trend | 0.08 | -0.33 | 0.49 | 0.69 | 0.98 |
| | UKESM Trend Forced | 0.46 | 0.09 | 0.84 | 0.01 | 0.99 |
| | UKESM+AKs Trend Forced | 0.10 | -0.32 | 0.51 | 0.64 | 0.98 |
| OMI – East Asia | Trend | -0.09 | -7.88 | 7.70 | 0.98 | 0.51 |
| | Trend Error 1 | -1.05 | -6.61 | 4.52 | 0.70 | 0.66 |
| | Trend Error 2 | 0.87 | -8.24 | 9.98 | 0.85 | 0.38 |
| | Apriori Trend | -0.25 | -0.71 | 0.22 | 0.29 | 0.98 |
| | UKESM Trend | -0.16 | -0.94 | 0.62 | 0.67 | 0.98 |
| | UKESM+AKs Trend | -0.62 | -2.24 | 1.00 | 0.44 | 0.95 |
| | UKESM Trend Forced | 0.90 | 0.34 | 1.47 | 0.00 | 0.99 |
| | UKESM+AKs Trend Forced | 1.02 | -0.04 | 2.09 | 0.05 | 0.97 |
| FORLI – East Asia | Trend | -1.52 | -2.16 | -0.88 | 0.00 | 0.93 |
| | Trend Error 1 | -1.42 | -2.06 | -0.78 | 0.00 | 0.93 |
| | Trend Error 2 | -1.62 | -2.27 | -0.98 | 0.00 | 0.92 |

| | | | | | |
|---|---|---|---|---|---|
| | Apriori Trend | -0.03 | -0.22 | 0.16 | 0.76 | 0.21 |
| | UKESM Trend | -0.03 | -0.62 | 0.56 | 0.93 | 0.98 |
| | UKESM+AKs Trend | -0.29 | -0.80 | 0.22 | 0.25 | 0.95 |
| | UKESM Trend Forced | 0.66 | 0.15 | 1.17 | 0.01 | 0.98 |
| | UKESM+AKs Trend Forced | 0.08 | -0.44 | 0.61 | 0.75 | 0.93 |
| SOFRID - East Asia | Trend | -0.19 | -1.01 | 0.63 | 0.65 | 0.96 |
| | Trend Error 1 | -0.08 | -0.73 | 0.58 | 0.82 | 0.90 |
| | Trend Error 2 | -0.30 | -1.02 | 0.42 | 0.41 | 0.93 |
| | Apriori Trend | -0.15 | -0.39 | 0.09 | 0.21 | 0.98 |
| | UKESM Trend | -0.42 | -0.97 | 0.13 | 0.12 | 0.99 |
| | UKESM+AKs Trend | -0.24 | -0.67 | 0.20 | 0.28 | 0.98 |
| | UKESM Trend Forced | 0.63 | 0.26 | 1.00 | 0.00 | 0.99 |
| | UKESM+AKs Trend Forced | 0.20 | -0.20 | 0.61 | 0.31 | 0.98 |

**Table 2:** *$LTCO_3$ trends (DU/decade) for the satellite trend (Trend), the satellite-uncertainty trend (Trend Error 1), the satellite+uncertainty trend (Trend Error 2), the satellite apriori trend (Apriori Trend), UKESM trend (UKESM Trend), UKESM with AKs applied trend (UKESM+AKs Trend), UKESM forced trend (UKESM Trend Forced) and UKESM with AKs applied forced trend (UKESM+AKs Trend Forced). The "trend lower" and "trend upper" represent the trend 95% confidence interval based on the trend precision calculated from **Equation 3**. $R^2$ is the trend fit skill (i.e. correlation squared) and the p-value is also shown.*

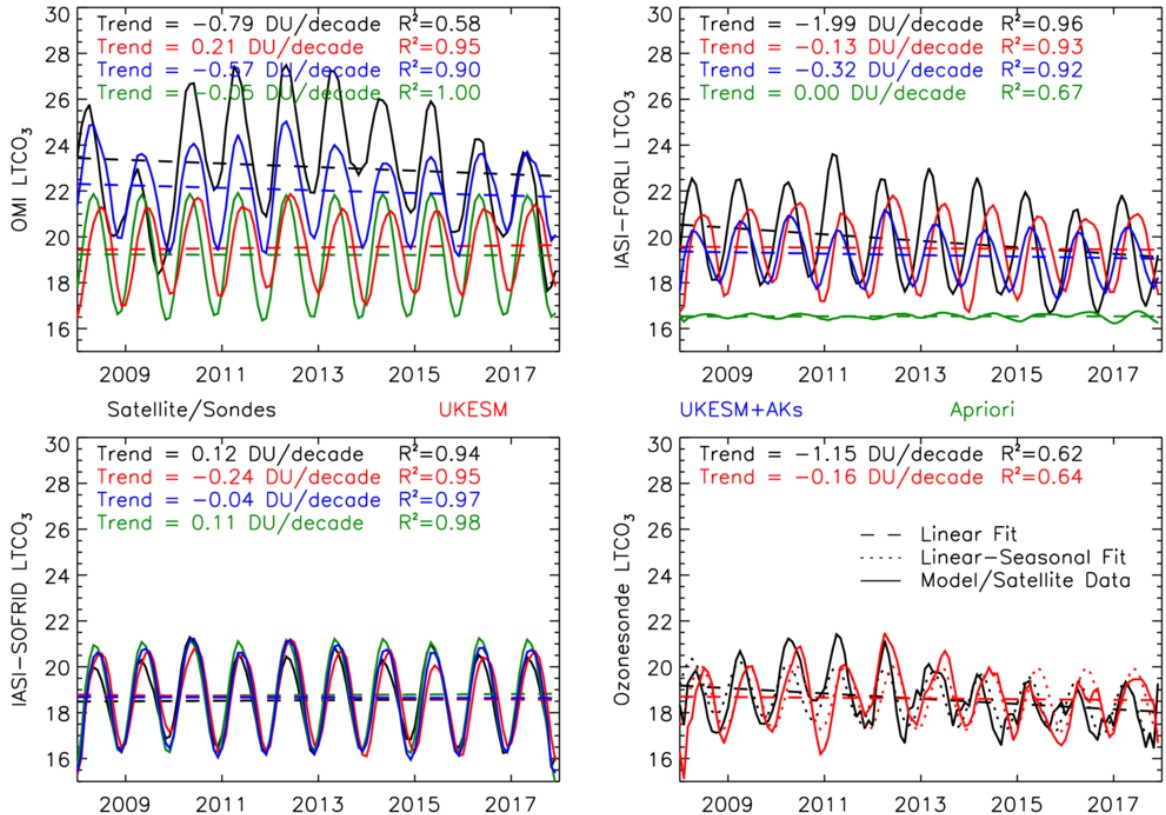

861

**Figure 1:** *Lower tropospheric column ozone (LTCO₃, surface to 450 hPa, DU) regional time-series for North America, based on the HTAP land mask, from OMI (top-left), IASI-FORLI (top-right), IASI-SOFRID (bottom-left) and ozonesondes (bottom-right) are shown by the black lines in the respective panels. UKESM simulations without and with satellite averaging kernels (AKs) applied are shown in red and blue lines. Green lines show the satellite apriori. Dashed lines show the LTCO₃ linear trend which are labelled in the top of each panel. The R² squared values show the linear-seasonal trend model fit to the corresponding LTCO₃ time-series (i.e. correlation squared).*

869

870

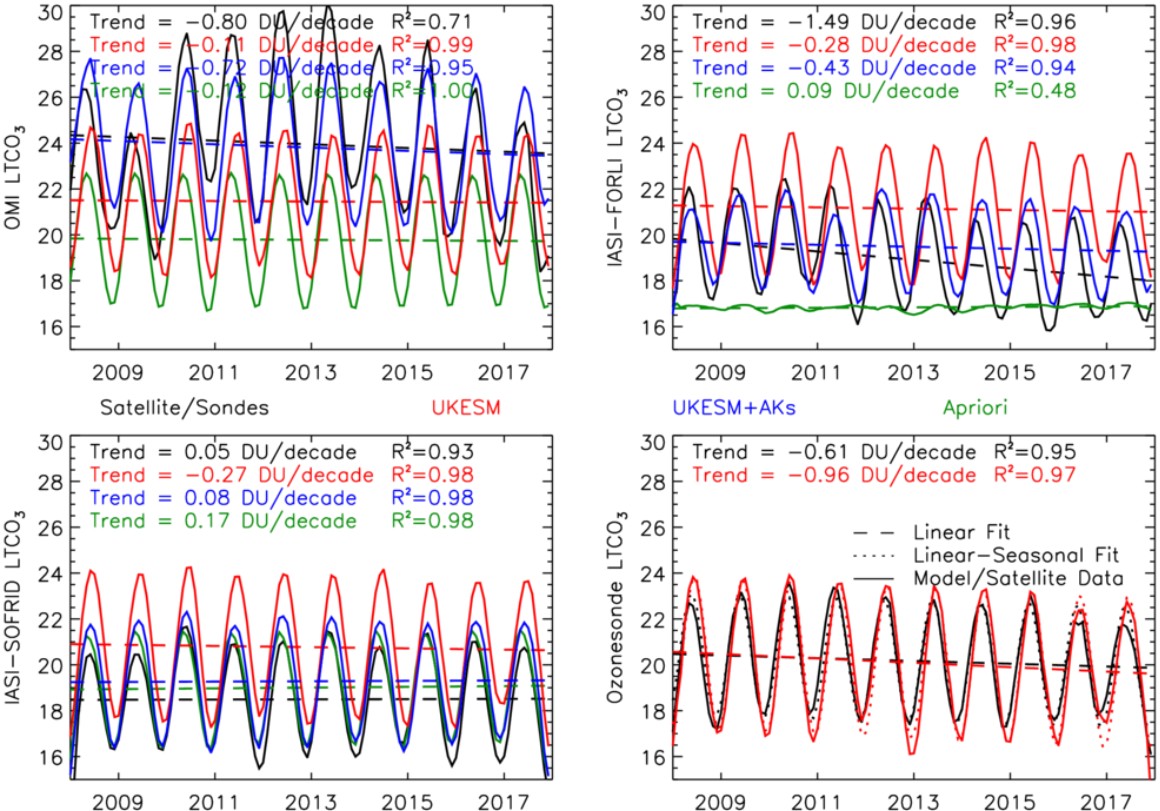

871

**Figure 2:** *LTCO₃ (DU) regional time-series for Europe, based on the HTAP land mask, from OMI (top-left), IASI-FORLI (top-right), IASI-SOFRID (bottom-left) and ozonesondes (bottom-right) are shown by the black lines in the respective panels.. UKESM simulations without and with satellite AKs applied are shown in red and blue lines. Green lines show the satellite apriori. Dashed lines show the LTCO₃ linear trend which are labelled in the top of each. The R² squared values show the linear-seasonal trend model fit to the corresponding LTCO₃ time-series (i.e. correlation squared).*

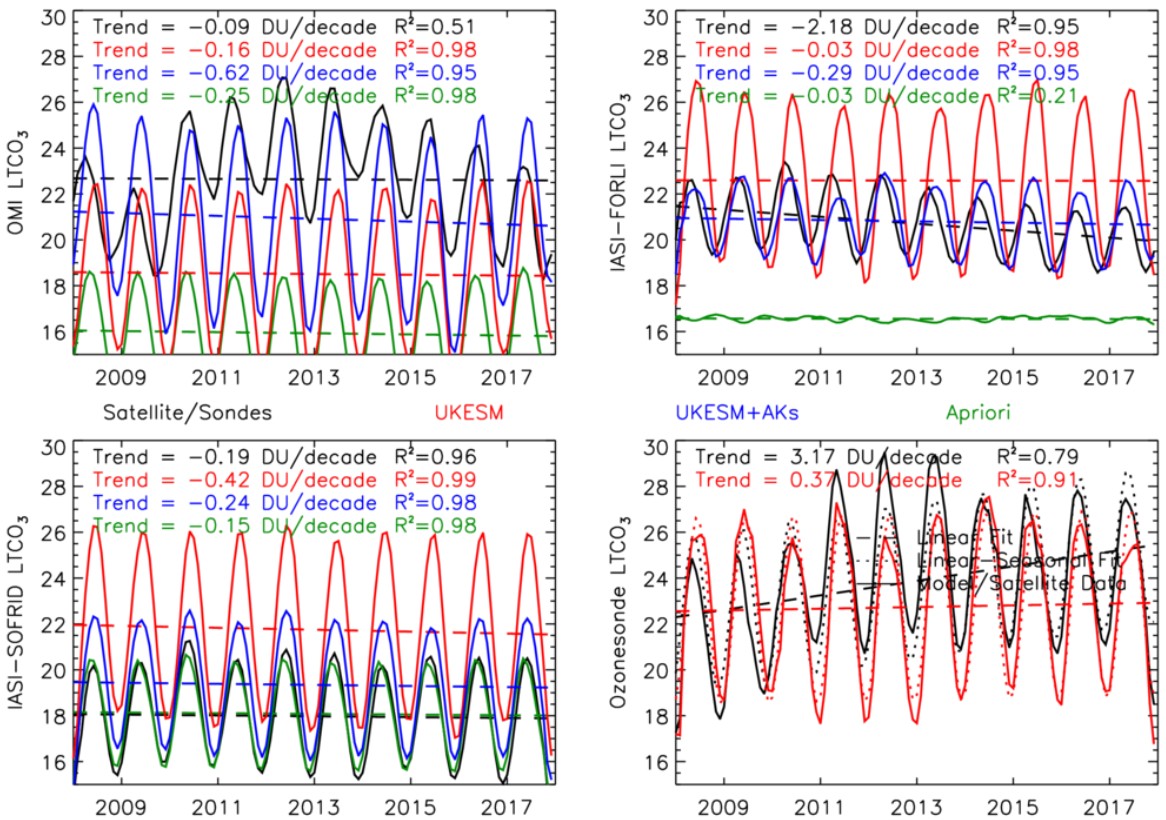

878

**Figure 3:** *LTCO₃ (DU) regional time-series for East Asia, based on the HTAP land mask, from OMI (top-left),*
*IASI-FORLI (top-right), IASI-SOFRID (bottom-left) and ozonesondes (bottom-right) are shown by the black lines*
*in the respective panels. UKESM simulations without and with satellite AKs applied are shown in red and blue*
*lines. Green lines show the satellite apriori. Dashed lines show the LTCO₃ linear trend which are labelled in the*
*top of each panel. The R² squared values show the linear-seasonal trend model fit to the corresponding LTCO₃*
*time-series (i.e. correlation squared).*

885

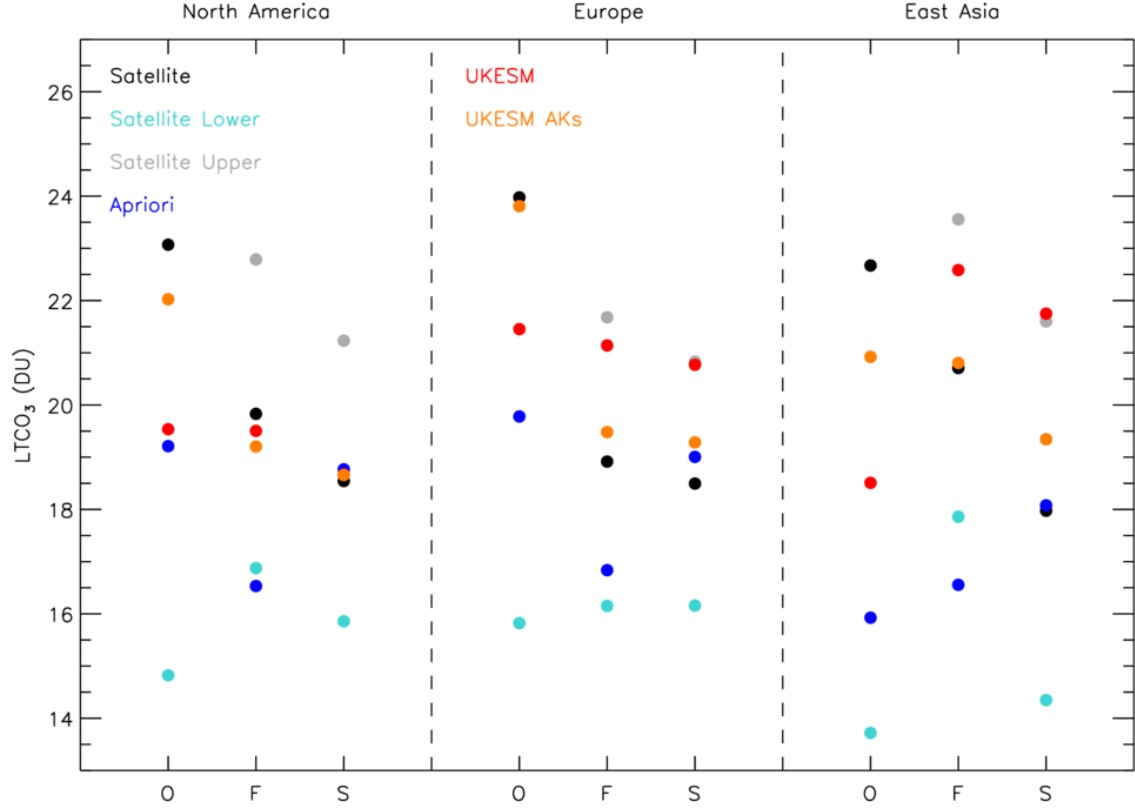

886

Figure 4: *Average LTCO$_3$ (DU) values across the 2008-2017 time-period for the satellite (black), satellite-lower (cyan), satellite-upper (grey), apriori (blue), UKESM (red) and UKESM+AKs (orange). The satellite-lower and satellite-upper values are the average of the satellite ± its error term time-series (note: these values do not always fit in the y-axis range). O, F and S represent OMI, IASI-FORLI and IASI-SOFRID for North America (left), Europe (centre) and East Asia (right).*

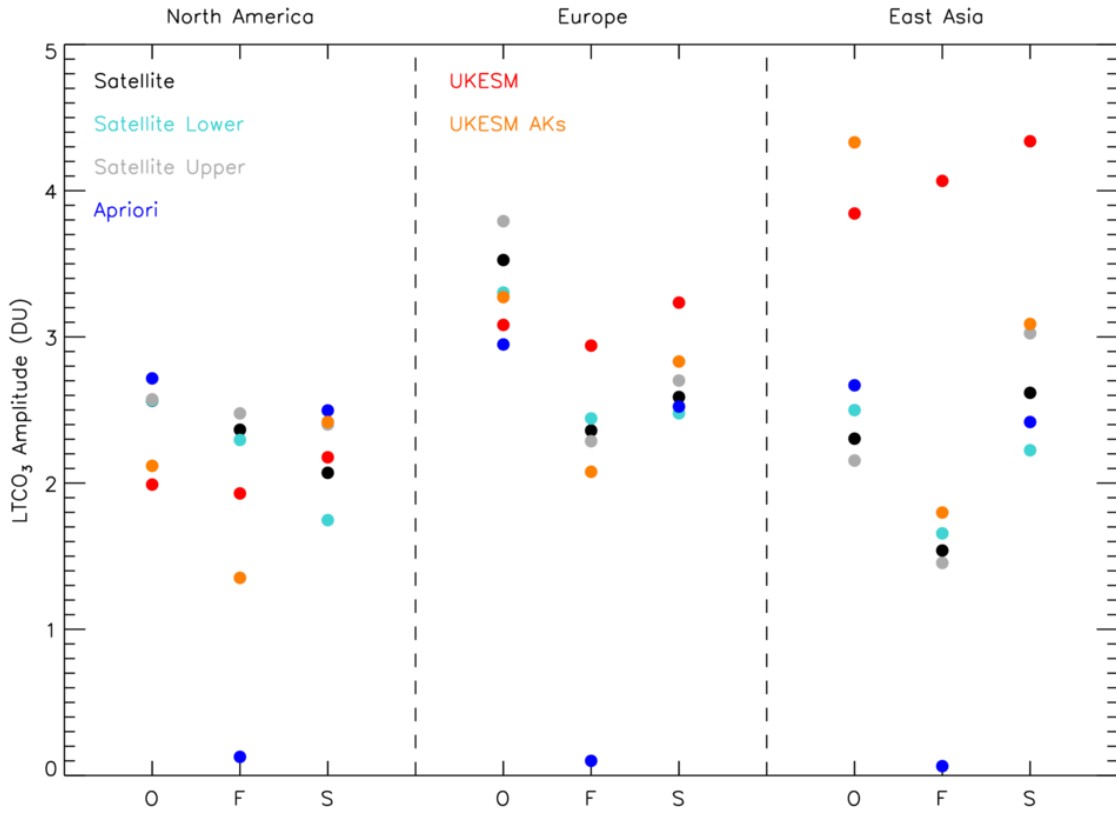

892

**Figure 5**: *Average LTCO$_3$ seasonal cycle amplitude (DU) values across the 2008-2017 time-period for the satellite (black), satellite-lower (cyan), satellite-upper (grey), apriori (blue), UKESM (red) and UKESM+AKs (orange). The satellite-lower and satellite-upper values are the average of the satellite ± its error term time-series (note: these values do not always fit in the y-axis range). O, F and S represent OMI, IASI-FORLI and IASI-SOFRID for North America (left), Europe (centre) and East Asia (right).*

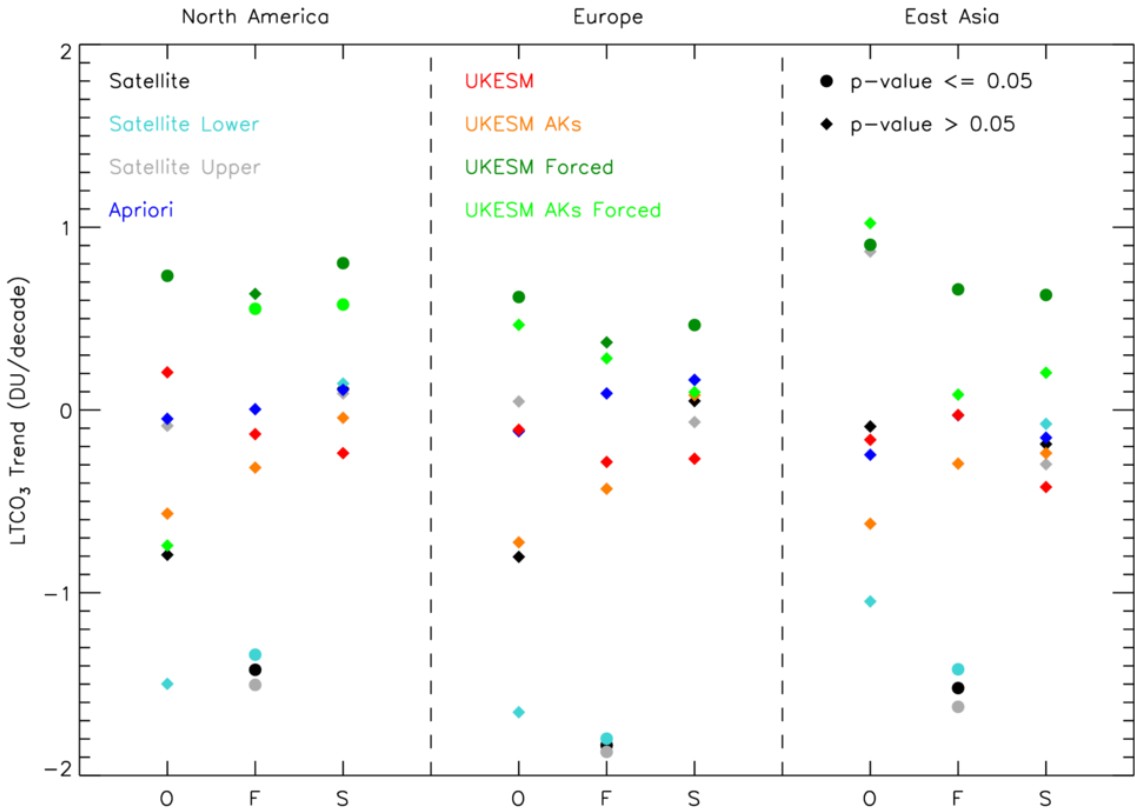


**Figure 6**: *Average LTCO₃ linear trends (DU/decade) values across the 2008-2017 time-period for the satellite (black), satellite-lower (cyan), satellite-upper (grey), apriori (blue), UKESM (red), UKESM+AKs (orange), UKESM forced (dark green) and UKESM+AKs forced (light green). The satellite-lower and satellite-upper values are the average of the satellite ± its error term time-series (note: these values do not always fit in the y-axis range). O, F and S represent OMI, IASI-FORLI and IASI-SOFRID for North America (left), Europe (centre) and East Asia (right). Triangle and circular symbols represent linear trends with p-values > 0.05 or p <= 0.05, respectively.*

