# Peer review of "Investigation of satellite vertical sensitivity on long-term retrieved lower tropospheric ozone trends 2 Richard J. Pope1,2, Fiona M. O'Connor3,4, Mohit Dalvi3, Brian J. Kerridge5,6, Richard Siddans5,6, Barry G. Latter5,6, Brice B"

_EGUsphere, 2023_

## Community Comment (CC1)

Comments by Owen R. Cooper (TOAR Scientific Coordinator of the Community Special Issue) on:

**Investigation of satellite vertical sensitivity on long-term retrieved lower tropospheric ozone trends**

Richard J. Pope, Fiona M. O'Connor, Mohit Dalvi, Brian J. Kerridge, Richard Siddans, Barry G. Latter, Brice Barret, Eric Le Flochmoen, Anne Boynard, Martyn P. Chipperfield, Wuhu Feng, Matilda A. Pimlott, Sandip S. Dhomse, Christian Retscher, Catherine Wespes, and Richard Rigby

This manuscript was submitted to ACP as part of the TOAR-II Community Special Issue
https://doi.org/10.5194/egusphere-2023-3109

This review is by Owen Cooper, TOAR Scientific Coordinator of the TOAR-II Community Special Issue. I, or a member of the TOAR-II Steering Committee, will post comments on all papers submitted to the TOAR-II Community Special Issue, which is an inter-journal special issue accommodating submissions to six Copernicus journals:  ACP (lead journal), AMT, GMD, ESSD, ASCMO and BG. The primary purpose of these reviews is to identify any discrepancies across the TOAR-II submissions, and to allow the author teams time to address the discrepancies.  Additional comments may be included with the reviews. While O. Cooper and members of the TOAR Steering Committee may post open comments on papers submitted to the TOAR-II Community Special Issue, they are not involved with the decision to accept or reject a paper for publication, which is entirely handled by the journal's editorial team.

**General Comments:**

TOAR-II has produced two guidance documents to help authors develop their manuscripts so that results can be consistently compared across the wide range of studies that will be written for the TOAR-II Community Special Issue.  Both guidance documents can be found on the TOAR-II webpage:
https://igacproject.org/activities/TOAR/TOAR-II

*The TOAR-II Community Special Issue Guidelines*:   In the spirit of collaboration and to allow TOAR-II findings to be directly comparable across publications, the TOAR-II Steering Committee has issued this set of guidelines regarding style, units, plotting scales, regional and tropospheric column comparisons, tropopause definitions and best statistical practices.

*The TOAR-II Recommendations for Statistical Analyses*:  The aim of this guidance note is to provide recommendations on best statistical practices and to ensure consistent communication of statistical analysis and associated uncertainty across TOAR publications. The scope includes approaches for reporting trends, a discussion of strengths and weaknesses of commonly used techniques, and calibrated language for the communication of uncertainty. Table 3 of the TOAR-II statistical guidelines provides calibrated language for describing trends and uncertainty, similar to the approach of IPCC, which allows trends to be discussed without having to use the problematic expression, "statistically significant".

**Major Comments:**
This analysis aims to investigate long-term ozone trends but it only focuses on 2008-2017, which is just 10 years.  Given the high interannual variability of ozone, the authors seem to conclude that 10 years is too short to investigate long-term trends.  Based on previous work, this finding is to be expected.  The following papers discuss the challenges of detecting ozone trends on short time scales, given the high degree of interannual variability, and limited sampling rates:  Barnes et al., 2016; Fiore et al., 2022;

Chang et al., 2020; Chang et al., 2024 (submitted to the TOAR-II Community Special Issue). It would be helpful if this topic can be discussed in the Introduction to set the stage for the analysis.

Figures 1, 2 and 3 compare satellite products to ozonesondes across fairly large regions. It's not clear how this comparison is done. It seems that the model is compared to co-located sondes, but what about the satellite products? Are the satellite products in these figures also compared to co-located sondes, or are the satellite products averaged over the full continental domains? The N. America, Europe and East Asia regions are very large (based on the HTAP map), but how many ozonesonde sites are available across these regions for comparison? A map showing the locations of the ozonesonde stations would be very helpful. I am aware that the western part of the N. America domain that stretches all the way to western Alaska has no in situ ozone observations west of Edmonton. Why include this far-flung region when it can't be compared to observations? Likewise, I don't know of any ozonesondes launched west of Beijing, yet the East Asia domain stretches all the way to Kyrgyzstan. The supplement describes the comparison of satellite products and ozonesondes for the purposes of deriving the BCFs. Sondes and satellite data are matched if they fall within 500 km of each other. Why is such a large distance permitted? This is probably fine if you are dealing with monthly means, but if you are matching sondes and satellite products for specific times (e.g. within 6 hours), this could introduce a lot of error. Previous work has shown that free tropospheric ozone can be highly variable on relatively small spatial scales of roughly 100 km, especially in the vicinity of fronts (Bethan et al., 1996; Cooper et a., 1998).

Figure S3 is not very informative as it gives no information regarding the correlation of the data. Gaudel et al. 2024 (submitted to the TOAR-II Community Special Issue) provide some very clear comparisons between observations (ozonesondes and IAGOS profiles) and a range of satellite products, based on co-located monthly means. These scatter plots show the bias and correlation and also show how the satellite products perform for the extreme values. It would be helpful if Figure S3 can be plotted in a similar way.

**Minor Comments:**

line 58
presents should be presence

line 64
Beginning a sentence with "However" suggests that this sentence contradicts the previous sentence, but it does not. This sentence merely reports the findings of Gaudel et al. 2018.

line 75
This sentence claims that the cause of the positive OMI trends and the negative IASI trends was caused by the sensitivity to different layers of the troposphere. But no study has ever shown this, and this was not a definitive conclusion of Gaudel et al. 2018.

In the conclusions, when mentioning future satellite programs, you can also list the NOAA GEO-XO and NEON satellite programs:
https://www.nesdis.noaa.gov/our-satellites/future-programs/geostationary-extended-observations-geoxo
https://www.nesdis.noaa.gov/our-satellites/future-programs/near-earth-orbit-network-neon

Figure S2
This plot is difficult to read because the colorbar is smooth rather than discrete. Please use discrete colors, and assign an HTAP regional number to each color.

**References**

Barnes, E.A., Fiore, A.M. and Horowitz, L.W.: Detection of trends in surface ozone in the presence of climate variability. Journal of Geophysical Research: Atmospheres, 121(10), 1099 pp.6112-6129, 2016.

Bethan, S., G. Vaughan, C. Gerbig, A. Volz-Thomas, H. Richer, and D. A. Tiddeman (1998), Chemical air mass differences near fronts, *J. Geophys. Res.*, 103(D11), 13413–13434, doi:10.1029/98JD00535.

Chang, K. L., Cooper, O. R., Gaudel, A., Petropavlovskikh, I., & Thouret, V.: Statistical regularization for trend detection: an integrated approach for detecting long-term trends from sparse tropospheric ozone profiles. Atmospheric Chemistry and Physics, 20(16), 9915-9938, 2020

Chang, K.-L., Cooper, O. R., Gaudel, A., Petropavlovskikh, I., Effertz, P., Morris, G., and McDonald, B. C.: Technical note: Challenges of detecting free tropospheric ozone trends in a sparsely sampled environment, EGUsphere [preprint], https://doi.org/10.5194/egusphere-2023-2739, 2024.

Cooper, O. R., J. L. Moody, J. C. Davenport, S. J. Oltmans, B. J. Johnson, X. Chen, P. B. Shepson, and J. T. Merrill, Influence of springtime weather systems on vertical ozone distributions over three North American sites, *J. Geophys. Res.*, *103*, 22,001-22,013. 1998.

Fiore, Arlene M., et al. (2022), Understanding recent tropospheric ozone trends in the context of large internal variability: A new perspective from chemistry-climate model ensembles, Environmental Research: Climate, https://doi.org/10.1088/2752-5295/ac9cc2

Gaudel, A., Bourgeois, I., Li, M., Chang, K.-L., Ziemke, J., Sauvage, B., Stauffer, R. M., Thompson, A. M., Kollonige, D. E., Smith, N., Hubert, D., Keppens, A., Cuesta, J., Heue, K.-P., Veefkind, P., Aikin, K., Peischl, J., Thompson, C. R., Ryerson, T. B., Frost, G. J., McDonald, B. C., and Cooper, O. R.: Tropical tropospheric ozone distribution and trends from in situ and satellite data, EGUsphere [preprint], https://doi.org/10.5194/egusphere-2023-3095, 2024.

---

## Author Comment (AC1)

**Author Responses to Reviewer Comments**

We thank the reviewers for their useful and constructive comments and feedback. We also thank Owen Cooper for his useful comments on our manuscript in relation to the TOAR-II special issue. We have reproduced their comments below in black text, followed by our responses in red text. Please note, we have number-listed Reviewer #1's and #2's and Owen Cooper's comments for clarification when addressing comments relevant to both reviewers. Any additions to the manuscript are in blue text and our references to line numbers are for the originally submitted manuscript.

**Reviewer 1's Comments:**

This work by Pope et al. has a promising title and research goal, but fails to meet its expectations. Large parts of the methodology are flawed or missing (see critical points), and the overall presentation is too vague or even sloppy (see comments on presentation). This work does not properly answer its research questions as a result. I cannot recommend it for publication in its present format.

We have responded to Reviewer #1's comments below, which address their overarching judgement in the paragraph above.

Critical points:

1. Why apply bias correction factors for trend studies? This is especially questionable if their construction involves 'application' of the averaging kernels that are under study! If this operation allows for more 'like-for-like comparisons' indeed, it should be rigorously and analytically explained in the main text, as a key methodology part of the paper.

Derivation of the bias correction factors (BCFs) aims to eliminate, so far as possible, systematic biases between the respective satellite records in order to harmonise the records. The BCFs do not change year-on-year so do not interfere with interannual variability within the individual records. This approach has been applied in both Pope et al. (2024) and Russo et al. (2023). This is discussed in S1 of the supplementary material (SM):

*"As satellite records can have systematic biases in column ozone (e.g. Gaudel et al., 2018), we use these BCFs in an attempt to harmonise the records in absolute value terms. Thus, as BCFs are generated from a long-term average, they should improve absolute column values, but not interfere with the long-term change in the record. This was done for lower tropospheric column ozone (LTCO$_3$), as discussed in the main manuscript."*

However, we agree that this point can be made clearer in the main manuscript, so on Page 3 Lines 115-117 we have updated:

*"To remove systematic biases between the satellite records, ozonesondes were used to generate bias correction factors (2008-2017) to help harmonise the data sets."* To:

"To remove systematic biases between the satellite records, while maintaining the long-term inter-annual variability of each record, ozonesondes were used to generate bias correction factors (2008-2017) to help harmonise the data sets (e.g. as done in Russo et al. (2023) and Pope et al. (2024)).".

Then on Page 3 Line 117, we have brought the discussion on the application of the AKs into the main manuscript:

"Here, each ozonesonde profile was co-located with the nearest satellite retrieval within 500 km and 6 hours to reduce spatiotemporal sampling biases (e.g. Keppens et al., 2019). The ozonesonde sonde profile was then interpolated in the vertical onto the satellite pressure grid where the sub-columns between pressure levels were determined. The ozonesonde sub-column profiles were then convolved

by the satellite averaging kernels (AKs), which represent the satellite's sensitivity to retrieval ozone as a function of altitude. Thus, allowing for a robust like-for-like comparison between the ozonesondes and the retrieved LTCO$_3$. The application of AKs to ozonesonde profiles to evaluate satellite ozone products is discussed in detail by Pope et al. (2023). The application of the AKs to the ozonesondes (and the model) is outlined in **Equation 1**:

$$sonde_{AK} = AK(sonde_{int} - apr) + apr \qquad (1)$$

where **sonde$_{AK}$** is the modified ozonesonde sub-column profile (Dobson units, DU), **AK** is the averaging kernel matrix, **sonde$_{int}$** is the sonde sub-column profile (DU) on the satellite pressure grid and **apr** is the apriori (DU). The application of the AKs to the ozonesondes is discussed in more detail in the Supplementary Material (SM) section **S1**.".

2. Why involve a model and apply the averaging kernels thereto (hence hiding certain information), even without mentioning how, if the retrieval sensitivity can be thoroughly studied and discussed in itself (see Rodgers, 2000)?

It is not completely clear what issue is being raised here. So, we would firstly point out that application of the AKs to model had been discussed in the SM, but we have now brought it into the main manuscript (see our response to Reviewer #1's Comment #1). Secondly, regarding why use a model? Models have often been used in studies to evaluate satellite data and the application of AKs to allow for like-for-like comparisons is a well-established practice (e.g. Pope et al., 2023, Pope et al., 2024 and Russo et al., 2023). In this work, the model allows for the assessment of how AKs can be linked to seasonal amplitudes and trends (or lack of). This was supported by Reviewer #2 with their comment "*it is still valuable to see the effects of the sampling and application of averaging kernels to the reference model*".

Now, while the AKs represent the sensitivity of the retrieval scheme as a function of altitude, we can take the trace of the AK matrix over the layers within the vertical range of this study (i.e. surface – 450 hPa), as in Pope et al. (2023), to derive the degrees of freedom of signal (DOFS), which quantify the pieces of independent information in LTCO$_3$. Therefore, to provide additional analysis of the retrieval vertical sensitivity and information content, we have derived the following **Figure S5** and new section for the SM **S3**: **Satellite Degrees of Freedom of Signal**:

[Figure]

**Figure S5:** *Satellite degrees of freedom of signal (DOFS) for the surface to 450 hPa layer regional time-series for North America (top-left), Europe (bottom-left) and East Asia (bottom-right). OMI, IASI-FORLI and IASI-SOFRID DOFS time-series are shown in red, blue and green, respectively. Dashed lines show the DOFS linear trend which are labelled in the top of each panel. The $R^2$ squared values show the linear-seasonal trend model fit to the corresponding DOFS time-series (i.e. correlation squared). The \* indicates where trends have a p-value < 0.05.*

**SM S3: Satellite Degrees of Freedom of Signal**

The degrees of freedom of signal (DOFS) represent the number of independent pieces of information from a satellite retrieval over a specified altitude range (e.g. total column, tropospheric column or lower tropospheric column (surface to 450 hPa) in this study). Here, we have used the satellite AKs to derive the $LTCO_3$ DOFS (i.e. trace of the AK matrix over the relevant satellite levels) and investigate how they have changed with time (**Figure S5**). For North America and Europe, the OMI and IASI-FORLI DOFS are approximately 0.5-0.7 and 0.3-0.5, respectively. However, for East Asia, their DOFS decrease to approximately 0.4-0.5 and 0.2-0.4, respectively. IASI-SOFRID typically has slightly lower DOFS, ranging between 0.2-0.4 across all three regions.

The long-term (2008-2017) trends in DOFS for all the products are relatively small ranging between -0.66 and 0.57 %/year. Only OMI shows substantial trends (i.e. p-value < 0.05) for North America and Europe. Overall, the $LTCO_3$ DOFS trends are small suggesting limited changes in the $LTCO_3$ information content of all the products and thus, unlikely to be contributing to the long-term $LTCO_3$ trends for the satellite records studies here.

In the main manuscript on Page 5 Line 204 we have added:

"We also investigated the satellite degrees of freedom of signal (DOFS) over the lower troposphere (i.e. surface to 450 hPa), which provides an estimate of the number of independent pieces of information in the $LTCO_3$. The DOFS are calculated by taking the trace of the AK matrix over the lower tropospheric levels in the satellite vertical grid. Overall, we found that the products for the three regions had negligible trends in their time-series (i.e. within ±1.0 %/year) meaning that the information content of satellite $LTCO_3$ had remained stable with time (see **S3**).".

On Page 6 Line 218:

"Interestingly, while the application of the IASI-FORLI AKs to UKESM marginally pushes the convolved model trend in LTCO$_3$ towards that of the satellite (which has a substantial negative trend), the IASI-FORLI DOFS actually have small positive trends (0.37-0.57 %/year – see **S3**). Therefore, there is minor scale, yet contrasting, discrepancy in how the vertical sensitivity is influencing the long-term LTCO$_3$ trends.".

We have also added a further short section to the paper to investigate the impact of diurnal variability on time-series of LTCO$_3$. While different satellites have different overpass times, it is difficult to disentangle the impact of the overpass time on the retrieved tropospheric ozone from satellite information alone. Here, the model adds a useful opportunity to explore this. When sampled at the overpass time rather than at UTC the only difference in the model-simulated LTCO$_3$ is that associated with diurnal variation. Therefore, we have added a new section (provisionally Section 3.3, but this will change in the final version once the suggestion of Reviewer #2 Comment #11 has been resolved):

**Section 3.3. Diurnal Variability on Regional LTCO$_3$ and Temporal Evolution**

As TO$_3$ varies diurnally due to meteorological and photochemical processes (e.g. Gaudel et., 2018), the different satellite overpass times (i.e. Aura and MetOp-A daytime overpasses are around 13:30 and 09:30 local time, respectively) will likely influence the spatial distributions of TO$_3$ which OMI and IASI will retrieve. In principle, this could therefore explain some differences between the two sensors and their long-term LTCO$_3$ trends. The model is a useful tool to help investigate this and we used the 6-hourly output to derived the UKESM simulated LTCO$_3$ spatial distributions at the Aura (13.30 LT) and MetOp-A (09.30 LT) day-time overpasses. These model fields were then used to calculate regional time-series for North America, Europe and East Asia. For each region and month, between 2008 and 2017, we calculated the regional average absolute difference (i.e. from the selection of model grid cells which fell within the HTAP-2 mask for a specific month) and the standard deviation of the absolute differences between the overpass times. Here, across all months and regions, we found the peak average absolute difference (13:30 LT – 09:30 LT) and standard deviation to be small at 2.03 and 2.56%, respectively. For the long-term trends, across all regions and overpass times, all of the UKESM trends were smaller than ±0.5 DU/decade. Therefore, the model LTCO$_3$ regional trends are negligibly different between overpass times. This might not be surprising given the negligible model trends in the satellite spatio-temporal trend comparisons (see **Section 3.1**), but the actual absolute differences (average and range) in simulated LTCO$_3$ are also small supporting the argument that on the regional scale, the day-time diurnal cycle differences between satellite overpass times has limited influence on the reported satellite trend discrepancies (e.g. in Gaudel et al., 2018).

3. The authors at several instances claim that the substantial IASI-FORLI trends are 'believed' to be due to changing meteorological input to the data processing in September 2010. The obvious check of doing independent trend studies before and after this change is missing.

There are several instances where we need to be more direct in our message about the IASI-FORLI discontinuity. Therefore, we have updated the text in the following places:

Page 5 Line 207: we have replaced "*though as suggested by Boynard et al., (2018) and Wespes et al., (2018)*" with "though as stated by Boynard et al., (2018) and Wespes et al., (2018)".

Page 6 Line 246: we have replaced "*again potentially attributable to*" with "again due to".

Page 10 Line 409-410: we have replaced "*that is believed to be due to changing meteorological input to the data processing.*" with "that is due to changing meteorological inputs to the data processing (Boynard et al., 2018; Wespes et al., 2018).".

Page 11 Line 443: we have replaced "*According to Boynard et al., (2018) and Wespes et al., (2018), the IASI-FORLI-v20151001 products*" with "As stated by Boynard et al., (2018) and Wespes et al., (2018), the IASI-FORLI-v20151001 products".

4. It is agreed with the authors that "Ideally, this analysis would have utilised several more records (e.g. several UV-Vis and IR products) to quantify long-term trends in LTCO3 and investigate the potential reasons for any discrepancies, as shown by Gaudel et al., (2018) for TCO3." At this point, none of these research goals is met. The authors could either focus on trend studies, or investigate whether the observed trends are (partially) due to spatio-temporal sensitivity changes. For the latter, it would suffice to compare ozone column trends with vertical sensitivity trends (without the need for models or bias correction).

We respectfully disagree with Reviewer #1 on their comments here.

In response to the comment "*At this point, none of these research goals is met*", we would contend that, although "null results" were obtained, such outcomes are of scientific value nonetheless. Indeed, as stated by Reviewer #2: "*This study presents a useful comparison of impacts from satellite sampling and vertical sensitivity on lower tropospheric ozone column regional averages, seasonal amplitudes and trends.*". We have presented a methodology to explore inconsistencies in LTCO$_3$ between several different satellite records which are currently available and conclude that, to fully achieve the stated scientific objective, further data and developments to the methodology will be required. Therefore, while more questions might have been raised from our study, this is valuable as it informs the further work needed to try and resolve this issue on tropospheric ozone trend inconsistencies. We have gone beyond the original study of Gaudel et al. (2018), who highlighted these issues in the first place, by employing an additional method (e.g. using a model as a tool to intercompare satellite vertical sensitivity impacts on trends) and metrics (e.g. DOFS, apriori, AKs and satellite uncertainties). Communicating these results in ACP will serve to inform fellow scientists of advances which will be necessary to address this issue in future studies.

In response to Reviewer #1's comment "*The authors could either focus on trend studies, or investigate whether the observed trends are (partially) due to spatio-temporal sensitivity changes. For the latter, it would suffice to compare ozone column trends with vertical sensitivity trends (without the need for models or bias correction).*", we have introduced a discussion on DOFS. "*Spatio-temporal sensitivity changes*" having already been addressed in part in the paper by deriving trends in apriori information. As stated in the manuscript, the apriori is for all three retrieval schemes and latitudes a trendless dataset and when deriving monthly means, where the spatio-temporal sampling of the level-2 swath data will differ between months, it shows limited long-term trends for all the data sets investigated here, thus the spatio-temporal sampling does not have a bearing on the satellite trend inconsistencies.

Comments on presentation:

5. First key point: "trends [...] show small scale trends" ?

We have updated the first key point from "*Satellite lower tropospheric column ozone (LTCO$_3$) trends in the northern hemisphere show small scale trends with large uncertainty ranges between 2008 and 2017.*" to "Satellite lower tropospheric column ozone (LTCO$_3$) records in the northern hemisphere show small trends with large uncertainty ranges between 2008 and 2017.".

6. Abstract: "year-to-year sampling is not an issue" is too vague. Do you mean changes in spatio-temporal sampling of satellite observations, or temporal changes in vertical smoothing (and

hence apriori contributions) of observations? I see this briefly explained between brackets in the discussion only.

To make this clearer, we have updated the Abstract on Page 1-2 Line 38-39 from "*Trends in the satellite a priori datasets show negligible trends indicating year-to-year sampling is not an issue*" to "Trends in the satellite apriori datasets show negligible trends indicating that any year-to-year changes in spatiotemporal sampling over of these satellite data sets over the period concerned has not influenced derived trends in LTCO$_3$.". However, we believe the general discussion is sufficient as we provide these statements:

Page 5 Line 190-191: "*The key point is that, as a climatology, the apriori will have no trend but if there are substantial temporal sampling differences between years, then an artificial trend could be introduced*".

Page 10 Line 405-407: "*Importantly, analysis of the three products' apriori LTCO3 records show negligible trends meaning that year-to-year sampling differences (i.e. the number of retrievals used in the spatial-monthly regional averages) are not skewing long-term satellite trends.*"

However, where we use the term "year-to-year sampling", we have replaced this with "year-to-year spatiotemporal sampling", to make this clearer for the reader.

7. In the introduction and discussion, the authors fail to acknowledge that an important reason for the discrepancies observed by Gaudel et al. (2018) was the use of different tropospheric top level definitions in different satellite products.

In the introduction, we have the paragraph:

"*The work by Gaudel et al. (2018) was part of the Tropospheric Ozone Assessment Report (TOAR), which represented a large global effort to understand spatio-temporal patterns and variability in TO$_3$. However, their investigation of ozonesondes (2003-2012) and products from nadir viewing satellites in polar orbits (three from the Ozone Monitoring Instrument (OMI) (2005-2015/6) and two from the Infrared Atmospheric Sounding Interferometer (IASI) (2008-2016)) displayed discrepancies in the spatial distribution, magnitude, direction and significance of the TCO$_3$ trends. They noted that the records cover slightly different time periods but were unable to provide any definitive reasons for these discrepancies beyond briefly suggesting that differences in measurement techniques and retrieval methods were likely to be causing the observed spatial inconsistencies.*"

In the paragraph, we state "*large global effort to understand spatio-temporal patterns and variability in TO$_3$*", so we are referring to Figure 24 of the Gaudel et al. (2018) paper which shows trends for 3 OMI and 2 IASI products. In Table 1, all of these products (i.e. the ones in Figure 24) use the WMO tropopause definition of the 2 K km$^{-1}$ lapse rate. So, in the paragraph above, they all use the same tropopause definition. However, to make it clear that some different definitions are used for other products (e.g. time-series analysis of SCIAMACHY data) we have added the following text on Page 2 Line 71:

"The range of potential definitions of the tropopause height used to derive TCO$_3$ from these nadir-viewing profile products could also lead to differences between the satellite product absolute values and their temporal evolution. While the 5 products discussed above use the same definition (i.e. World Meteorological Organisation (WMO) 2 K/km lapse rate; WMO, 1957), several of the other products analysed by Gaudel et al. (2018) did use other definitions.".

8. All apriori data should be properly introduced and discussed, given its relevance for the results interpretation (e.g. missing seasonality in IASI-FORLI apriori).

This is a good point and we have added the following text on Page 3 Line 108:

"For the ozone a priori profile, the RAL Space and FORLI schemes use the ozone latitude vs month of year climatology of McPeters et al. (2007), while IASI-SOFRID uses the dynamical ozone climatology described in Sofieva et al. (2014). However, the FORLI scheme uses a single ozone profile (Boynard et al., 2018) derived from the McPeters et al. (2007) dataset, so has no seasonality nor latitude dependence unlike the other retrieval schemes.".

9. What kind of averaging kernels are provided with the IASI-SOFRID L3 product?

The AKs from the IASI-SOFRID level-3 product are calculated from the gridded level-3 dataset (i.e. an AK per grid box). To make this clear, we have added the following text on Page 4 Line 137: "The satellite AKs from OMI and IASI-FORLI are provided in the level-2 files (i.e. an AK matrix per retrieval). However, the IASI-SOFRID AKs are provided from the gridded level-3 data product (i.e. an AK matrix for each 1°×1° grid box).".

10. C in equation 1 must represent an ozone value, not a month.

On Page 5 Line 168, we have replaced "*C is the first month of the record*" with "C is the first monthly mean LTCO$_3$ value of the record".

11. In sections 3.1 and 3.2, the essential information is lost in overdetailed number repetition that should be succinctly summarized. Correspondingly, Table 2 should go into the supplement

It is true that we provide a lot of numerical detail in these sections. This follows the TOAR-II special issue guidelines. For comparisons with other studies in TOAR and TOAR-II, authors are encouraged to include information on the trends, 95% confidence intervals and the p-values. We therefore believe it to be beneficial to keep Table 2 in the manuscript. A new table has been added to the SM with trends in units of ppbv/decade, to complement trends in DU/decade, the units used throughout the main paper. Please see also our response to Owen Cooper's Comment #2. Originally, in a draft before submission, we did not have Table 2 as the trend info was in the figures. However, as per the instructions from TOAR-II, we included Table 2 as we could not include e.g. the p-values in the figures. However, to try and make the text easier to read (i.e. not too many numbers), we have removed the p-values from the text as the reader can find these in Table 2, if needed.

12. Stating that "individual retrievals of LTCO3 are subject to multiple issues (…) which can result in noisy LTCO3 time-series" (lines 377-380) sounds unscientific, and diminishes the efforts done by satellite retrieval teams.

This statement is certainly not intended to "diminish" the efforts of satellite retrieval teams, but the wording has now been adjusted to ensure clarity on this point for readers. The text on Page 9 Lines 377-380 has been changed from:

"*Secondly, individual retrievals of LTCO$_3$ are subject to multiple issues (e.g. influences on radiative transfer and vertical sensitivity of stratospheric ozone, cloud and other particulates, surface spectral reflectivity/emissivity and temperature profile) which can result in noisy LTCO$_3$ time-series at high resolution (e.g. when gridded on a scale of 0.5° or 1.0°). Both of these factors supported analysis at a regional scale (e.g. continental scale).*" to

"Secondly, individual retrievals of LTCO$_3$ are often associated with large uncertainties (e.g. random and systematic uncertainties). There are multiple contributory factors concerning both instrumental

attributes (notably spectroradiometric noise and calibration accuracy) and variability in geophysical variables which influence radiative transfer and vertical sensitivity (e.g. stratospheric ozone, cloud and aerosol, surface spectral reflectivity/emissivity and pressure and temperature profile) which can result in $LTCO_3$ time-series with substantial variability/noise when derived at high spatial resolution (e.g. when deriving time-series from data gridded at 0.5° or 1.0°). Therefore, we undertake our analysis at the regional (e.g. continental) scale where more satellite retrievals are included in time-series monthly means yielding a reduction in the random error component of the sample.".

13. In the discussion, stating that "The IASI-SOFRID LTCO3 and apriori are very similar, with little inter-annual variability, which suggests that the IASI-SOFRID O3 retrieval in this height-range is more constrained by the apriori (i.e. less TO3 sensitivity than the other products)." is not necessarily true. The apriori can be close to the retrieved value, even for a perfect retrieval. Hence, again, the need for proper sensitivity studies, instead of derived quantities.

We refer to our response to Reviewer #1's Comment #2. On Page 10 Line 404, we have modified "*(i.e. less TO3 sensitivity than the other products)*" to "(i.e. less $TO_3$ sensitivity than the other products – see **S3**)".

14. Why are ozonesondes considered in 30° latitude bins, and not matched with the three regions under study? Moreover, it is not mentioned how many stations / launches are eventually involved.

For the ozonesonde-satellite corrections, we apply a consistent approach as outlined in Russo et al. (2023) and Pope et al. (2024). For the ozonesonde time-series shown in comparisons for Europe, North America and East Asia, only those which fall within these HTAP regions are selected. However, we appreciate that this is not clear in the main manuscript. Therefore, in Section 2.1 Page 3 Line 121, at the first mention of HTAP, we have added the following statement to make it clear how the ozonesondes are selected for the regional trends and how many ozonesonde sites there are in each.

"For the ozonesonde time-series for each HTAP region investigated, only ozonesonde sites which are located within each HTAP region are selected. This results in 15, 13 and 6 ozonesonde sites for North America, Europe and East Asia, respectively. As ozonesonde data for East Asia are all from Japan, Taiwan and Hong Kong, trends in ozone $LTCO_3$ will likely be different to satellite/model trends over all East Asia.".

15. How are data interpolated between different vertical representations?

This is a good point. The IASI data are a set of sub-columns between the surface and top of atmosphere. As this set of sub-columns is spaced more closely than the nominal 0-6 km $LTCO_3$ layer, the sub-columns are added from the surface to the tropopause. More information on this is in our response to Reviewer #1's Comment #17. For the model and ozonesondes, where they are not compared to the satellite data, the sub-columns are derived between levels and are totalled up in a similar method to the satellite data. In the case where the model and/or ozonesondes are compared to the satellite data, the model/sonde profile (units of mixing ratio) is interpolated in log(pressure) onto the satellite pressure grid, the sub-columns determined, the AKs applied and then the $LTCO_3$ derived. This has already been outlined in the SM, section S1:

"*Firstly, the co-located ozonesonde profile (in volume mixing ratio) is interpolated onto the satellite pressure grid in log($pressure$). The sonde sub-columns are then derived using the hydrostatic balance approximation...*"

Also, to make it clearer, we have split Section 2.1 into 3 separate paragraphs (Lines 96-106, Lines 107-115 and Lines 116-124).

16. Equations 2 and 3 are essentially the same. One should not make a difference based on satellite data formatting, which is irrelevant.

The reviewer is correct that Equations S2 and S3 are essentially the same. Therefore, we have removed Equation 2 from the SM and used Equation 3 for all the product AKs.

17. How are IASI sub-columns totalled up the 450 hPa level? Does this require interpolation between levels?

No, the IASI sub-columns are totalled up to the 450 hPa level. Where the 450 hPa level sits within the pressure range of a sub-column, the fraction between the bottom pressure level and the 450 hPa level is taken and used to scale this sub-column before being added to the sub-columns below (i.e. towards the surface). To make this clearer, we have added on Page 3 Line 115:

"The lowest sub-column in the OMI sub-column profile represents the surface-450 hPa layer (i.e. $LTCO_3$). For the IASI products, there were several sub-columns spanning the surface to 450 hPa range. Therefore, the IASI sub-columns were totalled up between the surface and the layer beneath or equal to the 450 hPa level. Where the 450 hPa level was located within a sub-column (i.e. was located between its bounding upper and lower pressure levels), the sub-column proportion between the lower pressure barrier and the 450 hPa level was determined and added to the sub-columns below (i.e. towards the surface).".

18. Figure S1: Why are correction factors (multiplicative) expressed in DU?

The BCFs are actually an offset, thus the units of DU. To make this clearer, we have changed the definition from BCFs to bias corrections offsets (BCOs) throughout the manuscript and updated the sentence on Page 3 Line 117 from:

*"This is discussed in the Supplementary Material (i.e. S1)."* to:

"Here, bias corrections offsets (BCOs) are systematic biases (i.e. subtraction term in units of Dobson units, DU) and is discussed in the Supplementary Material (i.e. S1).".

19. SM-3 should be part of the discussion (if somehow the model comparison is maintained).

The main purpose of using the model in our study is to aid interpretation of the long-term $LTCO_3$ trend and to help investigate the influence of the satellite AKs on several metrics (e.g. trends, time-series amplitude and absolute values). Please see our response to Reviewer #1's Comment #2 for more justification on using the model. Evaluation of the model included here is intended only to show that the model can simulate $LTCO_3$ sufficiently well to inform investigation of long-term satellite trends in the main manuscript. Therefore, we believe it makes sense to keep the model evaluation in the SM.

20. Not all authors are in the Author Contributions.

This has been updated to (note LJV has been removed as should not have been included initially):

"RJP conceptualised, planned and undertook the research study. BB, ELF, BJK, RS, BGL, AB and CW provided the OMI and IASI ozone data and advice on using the products and their analysis. FO and MD provided advice and expertise on using and running UKESM. CR provided advice and help during RP's ESA CCI fellowship. Scientific and technical contributions came from MPC, WF, MAP, SSD and RR. RJP prepared the manuscript with input from all co-authors.".

**Reviewer 2's Comments:**

This study presents a useful comparison of impacts from satellite sampling and vertical sensitivity on lower tropospheric ozone column regional averages, seasonal amplitudes and trends. Although the trend comparison from the IASI-FORLI data is impeded by known discontinuities in the record due to retrieval inputs, it is still valuable to see the effects of the sampling and application of averaging kernels to the reference model. I recommend publication after providing more information and references and addressing some structural and readability issues.

1. L 47: "other factors that require further investigation" Please be more specific here since TOAR-II is trying to address these issues.

We have added the following text to make this clearer:

"(e.g. the radiative transfer scheme (RTS) used and/or the inputs (e.g. meteorological fields) used in the RTS).".

2. L 73: "... will have an impact on which part of the troposphere the O3 signal is weighted towards" sounds somewhat conf–sing - maybe reword to:   "... will have an impact on the weighting of the tropospheric O3 signal in the reported satellite products"

We have reworded this text in-line with what Reviewer #2 has proposed.

3. L 79: "and around 11-12 km for IASI".  This is true for the maximum and in the global average, but neglects that there is sensitivity (i.e., a secondary peak)  in the lower troposphere around 5 km (e.g. Boynard et al. 2009)

This is a good point. Therefore, on P2 Line 79, we have added the following text ", while there is a secondary peak at approximately 5 km (Boynard et al., (2009).". We have added the reference provided by Reviewer #2 below to our reference list.

4. L 116: The supplementary material should also show the bias correction factors for the IASI data versions. Supplementary material could also discuss issues with temperature, water vapor input to IASI retrievals and how that potentially affects bias correction.

We have now included the BCF plots for IASI-FORLI and IASI-SOFRID shown below. We have also updated the text in the SM S1 from "*An example of the OMI LTCO$_3$ BCFs is shown in **Figure S1**.*" to "The OMI, IASI-FORLI and IASI-SOFRID LTCO$_3$ BCOs are shown in **Figure S1**, **Figure S2** and **Figure S3**, respectively.". Reviewer #2 also makes an interesting point about the influence of temperature and water vapour on the IASI retrievals and potentially therefore satellite-ozonesonde biases. While this would be an interesting and useful exercise in a future study, we do not consider it to be in scope or strictly necessary for this current study which aimed to quantify and remove biases so far as possible to derive the satellite LTCO$_3$ trends rather than to determine their causes.

[Figure]

**Figure S2**: *IASI-FORLI - ozonesonde (with AKs applied) BCOs (DU) for IASI-FORLI LTCO$_3$ using the instrument record between 2008 and 2017.*

[Figure]

**Figure S3**: *IASI-SOFRID - ozonesonde (with AKs applied) BCOs (DU) for IASI-SOFRID LTCO$_3$ using the instrument record between 2008 and 2017.*

5. L 116: This description of BCFs should mention that these are based on record monthly averages and would not change trends. This section should also include discussion of how this bias correction compares to the harmonization methods of Keppens et al., 2019.

We refer the reviewer to our response to Reviewer #1's Comment #1.

6. L 292: This section would be easier to read if the average ± error information was in a table

In Table 2 and the Section 3.2, we have presented the trend information as the trend, trend range – 95% confidence interval and the p-value. We have done this to meet the recommended criteria of the TOAR-II Special Issue guidelines. Therefore, we respectfully suggest that this information be left as it is. However, as in our response to Reviewer #1 Comment #11, we have removed the p-values (which can be retrieved from Table 2) in the text to reduce the occurrence of numbers in the text.

7. L 304: If you keep this in the text, I think ±error (DU) would be easier to interpret than error ranges

Pease see our response above to Reviewer #2's Comment #6.

8. L 379: should include water vapor

We have added "water vapour" to the variable list on Page 9 Line 379.

9. References in the supplementary material were not included there.

Good point. We have now included the following references in the SM and then also put the SM references (where they do not already exist) in the reference list of the main manuscript:

European Commission: Hemispheric Transport Air Pollution (HTAP): Specification of the HTAP2 experiments. https://ec.europa.eu/jrc (last accessed 15/05/2024), 2016.

Koffi, B., Dentener, F., Janssens-Maenhout, G., Guizzardi, D., Crippa, M., Diehl, T. and Galmarini, S. 2016.

Gaudel, A., Cooper, O.R., Ancellet, G., Barret, B., Boynard, A., Burrows, J.P., Clerbaux, C., Coheur, P.F., Cuesta, J., Cuevas, E., Doniki, S., Dufour, G., Ebojie, F., Foret, G., Garia, O., Granados-Munoz, M.J., Hannigan, J.W., Hase, F., Hassler, B., Huang, G., Hurtmans, D., Jaffe, D., Jones, N., Kalabokas, P., Kerridge, B., Kulwaik, S., Latter, B., Leblanc, T., Le Flochmoen, E., Lin, W., Liu, J., Liu, X., Mahieu, E., McClure-Begley, A., Neu, J.L., Osman, M., Palm, M., Petetin, H., Petropavlovskikh, I., Querel, R., Rahpoe, N., Rozanov, A., Schultz, M.G., Schwab, J., Siddans, R., Smale, D., Steinbacher, M., Tanimoto, H., Tarasick, D.W., Thouret, V., Thompson, A.M., Trickl, T., Weatherhead, E., Wespes, C., Worden, H.M., Vigouroux, C., Xu, X., Zeng, G. and Ziemke, J..: Tropospheric Ozone Assessment Report: Present day distribution and trends of tropospheric ozone relevant to climate and global atmospheric chemistry model evaluation. *Elementa*, 6(39), 1-58, doi:10.1525/elementa.291, 2018.

Hubert, D., Lambert, J-C., Verhoelst, T., Granville, J., Keppens, A., Baray, J-L., Bourassa, A.E., Cortesi, U., Degenstein, D.A., Froidevaux, L., Godin-Beekmann, S., Hoppel, K.W., Johnson, B.L., Kyrola, E., Leblanc, T., Lichtenberg, G., Marchand, M., McElroy, C.T., Murtagh, D., Nakane, H., Portafaix, T., Querel, R., Russell, J.M., Salvador, J., Smit, H.G.J., Stebel, K., Steinbrecht, W., Strawbridge, K.B., Stubi, R., Swart, D.P.J., Taha, G., Tarasick, D.W., Thompson, A.M., Urban, J., van Gijsel, J.A.E., Van Malderen, R., von der Gathen P., Walker, K.A., Wolfram, E. and Zawodny, J.M.: Ground-based assessment of the bias and long-term stability of 14 limb and occultation ozone profile data records. *Atmospheric Measurement Techniques*, 9, 2497-2534, doi: 10.5194/amt-9-2497-2016, 2016.

Keppens, A., Lambert, J-C., Graville, J., Hubert, D., Verhoelst, T., Compernolle, S., Latter, B., Kerridge, B., Siddans, R., Boynard, A., Hadji-Lazaro, J., Clerbaux, C., Wespes, C., Hurtmans, D.R., Coheur, P-F., van Peet, J.C.A., van der A, R.J., Garane, K., Koukouli, M.E., Balis, D.S., Delcloo, A., Kivi, R., Stubi, R., Godin-

Beekmann, S., Van Roozendael, M. and Zehner, C.: Quality assessment of the Ozone_cci Climate Research Data Package (release 2017) – Part 2: Ground-based validation of nadir ozone profile data products. *Atmospheric Measurement Techniques*, 11, 3769-3800, doi: 10.5194/amt-11-3769-2018, 2018.

10. L 58: presents -> presence

This has been updated.

11. L 187 titled subsections for each region (N. America, etc.) would help the reader here.

We agree with the reviewer, and we have now restructured the text in Section 3 with sub-sections for each geographic region.

12. L 218 start new paragraph for IASI-SOFRID results

We have now started a new paragraph in the text on Line 218.

13. L 220 also new paragraph for ozonesondes

We have now started a new paragraph on Line 224 (Line 220 is still IASI-SOFRID discussion).

References needed:

There reference have been added to the reference list.

Boynard, A., Clerbaux, C., Coheur, P.-F., Hurtmans, D., Turquety, S., George, M., Hadji-Lazaro, J., Keim, C., and Meyer-Arnek, J.: Measurements of total and tropospheric ozone from IASI: comparison with correlative satellite, ground-based and ozonesonde observations, Atmos. Chem. Phys., 9, 6255–6271, https://doi.org/10.5194/acp-9-6255-2009, 2009.

Keppens, A., Compernolle, S., Verhoelst, T., Hubert, D., and Lambert, J.-C.: Harmonization and comparison of vertically resolved atmospheric state observations: methods, effects, and uncertainty budget, Atmos. Meas. Tech., 12, 4379–4391, https://doi.org/10.5194/amt-12-4379-2019, 2019.

**Owen Cooper's Comments:**

General Comments:

1. TOAR-II has produced two guidance documents to help authors develop their manuscripts so that results can be consistently compared across the wide range of studies that will be written for the TOARII Community Special Issue. Both guidance documents can be found on the TOAR-II webpage: https://igacproject.org/activities/TOAR/TOAR-II

Thank you for pointing us towards these resources. As in our previous two manuscripts in the TOAR-II special issue, we have adhered to the advice for the majority of our work in this study as well.

2. The TOAR-II Community Special Issue Guidelines: In the spirit of collaboration and to allow TOAR-II findings to be directly comparable across publications, the TOAR-II Steering Committee has issued this set of guidelines regarding style, units, plotting scales, regional and tropospheric column comparisons, tropopause definitions and best statistical practices.

As in our previous two manuscripts in this TOAR-II special issue, we believe we have complied with the majority of the TOAR-II guidelines. We do acknowledge though that the trends in Table 2 are in units of DU/decade and not in ppbv/decade. Therefore, we have added an additional version of the Table with these units in the SM. We have also removed the colouring of the table to show different regions and used titles instead. The table with units of ppbv/decade is shown at the bottom of this response document. The updated Table 2 with no colouring is also shown in-line with the ACP guidelines.

3.  The TOAR-II Recommendations for Statistical Analyses: The aim of this guidance note is to provide recommendations on best statistical practices and to ensure consistent communication of statistical analysis and associated uncertainty across TOAR publications. The scope includes approaches for reporting trends, a discussion of strengths and weaknesses of commonly used techniques, and calibrated language for the communication of uncertainty. Table 3 of the TOAR-II statistical guidelines provides calibrated language for describing trends and uncertainty, similar to the approach of IPCC, which allows trends to be discussed without having to use the problematic expression, "statistically significant".

We have done our best to follow the TOAR-II recommendations for "Statistical Analysis". For instance, we provide the trend, 95% confidence range and the corresponding p-values. Regarding the units for the trends, as in our response to Owen Cooper's Comment #2, we have added a new table to the SM in units of ppbv/decade to complement our analysis in DU/decade.

Major Comments:

4.  This analysis aims to investigate long-term ozone trends but it only focuses on 2008-2017, which is just 10 years. Given the high interannual variability of ozone, the authors seem to conclude that 10 years is too short to investigate long-term trends. Based on previous work, this finding is to be expected. The following papers discuss the challenges of detecting ozone trends on short time scales, given the high degree of interannual variability, and limited sampling rates: Barnes et al., 2016; Fiore et al., 2022; Chang et al., 2020; Chang et al., 2024 (submitted to the TOAR-II Community Special Issue). It would be helpful if this topic can be discussed in the Introduction to set the stage for the analysis.

In the manuscript Page 10 Lines 382-393, we state that it would be beneficial for more records to be used and over a substantially longer time-period. However, IASI was first launched on MetOp-A in late 2006 and, at the time of undertaking this study, 2008-2017 was the only period for which IASI full-year data records were available from which to derive time-series of LTCO$_3$. The IASI-SOFRID data product has only recently been extended and the IASI-FORLI data is currently undergoing a reprocessing with consistent level-2 EUMETSAT meteorological information. In addition, although OMI was launched in 2004, the RAL Space data set is not due to be re-processed and extended to the present until later this year.

However, it is of course valid to state that inter-annual variability will have a more substantial impact on the linear trend determined from the record of LTCO$_3$ or TCO$_3$ from a single decade than multiple decades. Therefore, on Page 3 Line 85 we have added:

"The determination of the linear trend in a satellite TCO$_3$ record(s) can also be difficult as many factors (e.g. chemistry, emissions, deposition and transport) control ozone interannual variability, especially on time-periods of a decade or less (Barnes et al., 2016; Change et al., 2020; Fiore et al., 2022).".

The Barnes et al., 2016; Change et al., 2020; Fiore et al., 2022 references have been added to the reference list.

5.  Figures 1, 2 and 3 compare satellite products to ozonesondes across fairly large regions. It's not clear how this comparison is done. It seems that the model is compared to co-located sondes, but what about the satellite products? Are the satellite products in these figures also compared to co-located sondes, or are the satellite products averaged over the full continental domains? The N. America, Europe and East Asia regions are very large (based on the HTAP map), but how many ozonesonde sites are available across these regions for comparison? A map showing the locations of the ozonesonde stations would be very helpful. I am aware that the western part of

the N. America domain that stretches all the way to western Alaska has no in situ ozone observations west of Edmonton. Why include this far-flung region when it can't be compared to observations? Likewise, I don't know of any ozonesondes launched west of Beijing, yet the East Asia domain stretches all the way to Kyrgyzstan. The supplement describes the comparison of satellite products and ozonesondes for the purposes of deriving the BCFs. Sondes and satellite data are matched if they fall within 500 km of each other. Why is such a large distance permitted? This is probably fine if you are dealing with monthly means, but if you are matching sondes and satellite products for specific times (e.g. within 6 hours), this could introduce a lot of error. Previous work has shown that free tropospheric ozone can be highly variable on relatively small spatial scales of roughly 100 km, especially in the vicinity of fronts (Bethan et al., 1996; Cooper et a., 1998).

Below we have repeated individual comments from Owen Cooper's Comment #5 and responded to them separately.

*Figures 1, 2 and 3 compare satellite products to ozonesondes across fairly large regions. It's not clear how this comparison is done. It seems that the model is compared to co-located sondes, but what about the satellite products?*

Yes, it is correct that the model and ozonesondes are co-located spatially and the trends determined for the ozonesonde sites in these HTAP regions as outlined on Page 6 Lines 224-229. As for the satellite data, the full spatial distribution over the HTAP regions is used. The BCFs are derived on a lat-month basis and duplicated longitudinally to be applied to the gridded satellite data products. We have made this clear by rewording the text on Page 3 Lines 118-121 from:

"*To investigate long-term trends over North America, Europe and East Asia, the Hemispheric Transport of Air Pollution (HTAP) regional sea-land mask (European Commission (2016); see S2, Figure S2), is used to generate average monthly time-series for each product over each region of interest.*" to:

"To investigate long-term trends over North America, Europe and East Asia, the Hemispheric Transport of Air Pollution (HTAP) regional sea-land mask (European Commission (2016); see S2, Figure S2), is used to sub-sample the gridded satellite data for the respective regions and then generate average monthly time-series for each product over each region of interest.".

*Are the satellite products in these figures also compared to co-located sondes, or are the satellite products averaged over the full continental domains? The N. America, Europe and East Asia regions are very large (based on the HTAP map), but how many ozonesonde sites are available across these regions for comparison?*

The ozonesondes are only used in conjunction with the satellite data to derive the latitude-month BCFs (more information in our response to Reviewer #1's Comments #1). These BCFs are then mapped onto the horizontal grid of the gridded satellite data (as demonstrated in Figure S1 and the section S1 of the SM). The actual discussion of the BCFs and how they are applied has been updated in our response to Reviewer #1's Comment #1 and Reviewer #2's Comment #4.

*A map showing the locations of the ozonesonde stations would be very helpful. I am aware that the western part of the N. America domain that stretches all the way to western Alaska has no in situ ozone observations west of Edmonton. Why include this far-flung region when it can't be compared to observations? Likewise, I don't know of any ozonesondes launched west of Beijing, yet the East Asia domain stretches all the way to Kyrgyzstan.*

We have now included the following map (new **Figure S1**) in the SM section S1 showing the location of the ozonesonde sites used for the trend analysis in Section 3 of the main manuscript.

[Figure]

**Figure S1:** *Locations of the ozonesondes used for deriving the bias correction offsets (BCOs), evaluating UKESM and calculating regional lower tropospheric column ozone (LTCO₃) trends (Dobson units, DU). The red, green and blue ozonesonde sites were used for deriving the regional long-term time-series (2008-2017) for North America, Europe and East Asia, respectively (see **S2** for more details on region definitions). Note, several ozonesonde sites will have overlapping circles.*

In the SM S1 (first paragraph), we have added the following sentence:

"The ozonesonde locations are shown in **Figure S1**".

*The supplement describes the comparison of satellite products and ozonesondes for the purposes of deriving the BCFs. Sondes and satellite data are matched if they fall within 500 km of each other. Why is such a large distance permitted? This is probably fine if you are dealing with monthly means, but if you are matching sondes and satellite products for specific times (e.g. within 6 hours), this could introduce a lot of error. Previous work has shown that free tropospheric ozone can be highly variable on relatively small spatial scales of roughly 100 km, especially in the vicinity of fronts (Bethan et al., 1996; Cooper et a., 1998).*

We have provided an updated discussion on the BCFs, which is relevant here, in-line with our response to Reviewer #1's Comment #1 and Reviewer #2's Comment #4. This approach has been used by both Russo et al. (2023) and Pope et al. (2024).

6. Figure S3 is not very informative as it gives no information regarding the correlation of the data. Gaudel et al. 2024 (submitted to the TOAR-II Community Special Issue) provide some very clear comparisons between observations (ozonesondes and IAGOS profiles) and a range of satellite

products, based on colocated monthly means. These scatter plots show the bias and correlation and also show how the satellite products perform for the extreme values. It would be helpful if Figure S3 can be plotted in a similar way.

In line with Owen Cooper's comment, we have updated Figure S3 to scatter plots (now Figure S6) which include useful metrics such as the correlation and percentage mean bias. The new figure is below:

[Figure]

**Figure S6**: *Comparison of UKESM and ozonesonde LTCO$_3$ (DU) between 2008 and 2017 for December-January-February (DJF), March-April-May (MAM), June-July-August (JJA) and September-October-November (SON) across the latitude bands: 90-30°S, 30°S-30°N and 30-90°N. The correlation R and percentage mean bias (MB%) metrics are shown for each panel.*

We have also updated the discussion section:

"For comparison with the ozonesondes (**Figure S6**), the model was co-located in time (within 6 hours) and space (nearest model grid box) with each of the ozonesondes. The analysis has been split up into three latitude ranges (90-30°S, 30°S-30°N & 30-90°N) and four seasons (December-January-February (DJF), March-April-May (MAM), June-July-August (JJA) and September-October-November (SON)). In the northern hemisphere (30-90°N), correlations range between 0.67 and 0.76 across the seasons with a relatively small percentage mean bias (MB%) of -12.94% to 4.39%. In the tropics (30°S-30°N), correlations range from 0.52 to 0.7 across the seasons with larger MB% values of 15.38% to 21.98%. The southern hemisphere (90-30°S) exhibits the strongest correlations between the model and observations, ranging between 0.72 and 0.90. The model underestimates LTCO$_3$, with MB% values of -22.28% to -8.79%. Overall, UKESM reproduces reasonably well the latitudinal-seasonal variations recorded in the ozonesondes for LTCO$_3$.".

Minor Comments:

7. line 58 presents should be presence

This has been updated.

8. line 64 Beginning a sentence with "However" suggests that this sentence contradicts the previous sentence, but it does not. This sentence merely reports the findings of Gaudel et al. 2018.

We have now removed "However" from the sentence in question.

9. line 75 This sentence claims that the cause of the positive OMI trends and the negative IASI trends was caused by the sensitivity to different layers of the troposphere. But no study has ever shown this, and this was not a definitive conclusion of Gaudel et al. 2018.

We concur and have reworded the sentence on Page 2 Line 75 "*This was evident in the OMI and IASI TCO$_3$ trends, where OMI showed predominantly positive trends between 60°S and 60°N while the opposite was the case for IASI*." to:

"This is potentially one of the drivers behind the different OMI and IASI TCO$_3$ trends, where OMI showed predominantly positive trends between 60°S and 60°N while the opposite was the case for IASI.".

10. In the conclusions, when mentioning future satellite programs, you can also list the NOAA GEO-XO and NEON satellite programs: https://www.nesdis.noaa.gov/our-satellites/future-programs/geostationary-extended-observationsgeoxo https://www.nesdis.noaa.gov/our-satellites/future-programs/near-earth-orbit-network-neo.

We have updated the final paragraph of the discussion Page 12 Lines 490-496 accordingly:

"In the near future, the new European polar orbiting mission MetOp Second Generation will include IASI Next Generation and Sentinel-5 UV/VIS sounders to provide height-resolved ozone products to extend current missions through to the mid-2040s. This will be supplemented by the new USA Near Earth Orbit Network (NEON) series as a replacement for the Joint Polar Satellite System (JPSS). The Geostationary Environment Monitoring Spectrometer (GEMS) and Tropospheric Emissions: Monitoring of Pollution (TEMPO) have also recently been launched and there will be new geostationary platforms: the Infrared Sounder (IRS) and Sentinel-4 UV/VIS sounder on Europe's Meteosat-Third Generation (MTG-S), again through to the mid-2040s, and the USA Geostationary Extended Observations (GeoXO) series. Overall, these platforms will provide large volumes of data (e.g. diurnal observations) and over a long-time scale on tropospheric ozone for future regional trend analyses.".

11. Figure S2 This plot is difficult to read because the colorbar is smooth rather than discrete colors, and assign an HTAP regional number to each color.

We have updated the HTAP figure using a discrete colour bar below:

[Figure]

**Figure S2**: *Mask of different regions provided by the Task Force on Hemispheric Transport of Air Pollution (HTAP) on a 1°×1° horizontal resolution. For instance, mask code 4 represents Europe.*

**References:**

Barnes, E.A., Fiore, A.M. and Horowitz, L.W.: Detection of trends in surface ozone in the presence of climate variability. Journal of Geophysical Research: Atmospheres, 121(10), 1099 pp.6112-6129, 2016.

[revised manuscript text omitted]

---

## Author Response (AR2)

**Author Response to Reviewer #1**

We thank Reviewer #1 for the additional comments. We have now addressed those below. The original reviewer comments are reproduced in black text, our responses in red text and additions to the manuscript in blue text. The reference to line numbers is based on the resubmitted manuscript and we have numbered Reviewer #1's comments.

1. Although I believe that the scientific quality and presentation of this work can still be improved (possibly requiring more time and resources than currently available), at this point I agree with the other reviewers that it has sufficient merit for inclusion in the TOAR-II special issue. The authors have corrected many initially poor phrasings and have substantially enhanced the manuscript content, especially on the methodology. Some questions remain, I believe, but they are no longer detrimental to publishing. The text still has a few erroneous phrasings and typos that should be corrected for.

We thank Reviewer #1 for their comments, and we are pleased to see that they are generally satisfied with our responses and update to the manuscript in the "Interactive Discussion" stage. We have now updated the manuscript to address the remaining concerns of the reviewer.

2. The update in reply to critical point 1 is appreciated, but for linear regressions Y = C + B*X (+ period + residuals, as in your Eq. 2), it is still not clear from the text (to me) why you would bother correcting for differences in the intercepts C if you are eventually only interested in the coefficients B (Table 2), especially if the corrections involve information that affects B? (cf. Reviewer 2, comment 5) Note: Critical points 2 and 3 have been extensively and appropriately addressed, respectively. The reply on the former at least partially addresses point 4 as well.

The reviewer is correct that the bias correction offsets (BCOs) have negligible influence on the linear trend (B) and correct for the systematic difference in the intercept (C). However, we believe that this is an appropriate step when presenting the data. By applying the BCOs, this creates a more robust product, which are then important for the evaluation of UKESM in the Supplementary Material (SM). To make this point clear, we have added the following text on Page 4 Line 139:

"By applying the BCOs, this is improving the robustness of the satellite datasets (in absolute terms). This is important when intercomparing the products but also when using them to evaluate UKESM and determining the model's skill to simulate $LTCO_3$ as used in this study (see **S4**).".

3. Lines 40-42: "Trends in the satellite apriori datasets show negligible trends indicating that any year-to-year changes in spatiotemporal sampling over of these satellite data sets over the period concerned has not influenced derived trends in." The erroneous phrasing that has been corrected for in the "key points" is still in the abstract: "trends … show … trends"

We have now reworded this on Page 1 & 2 Lines 36-38:

"The satellite apriori datasets show negligible trends indicating that any year-to-year changes in spatiotemporal sampling of these satellite data sets, over the period concerned, has not artificially influenced their $LTCO_3$ temporal evolution.".

4. Line 151: "ozonesonde sonde"

This has been corrected.